# REFOCUS: Visual Editing as a Chain of Thought for Structured Image Understanding

**Xingyu Fu** [* 1]  **Minqian Liu** [2]  **Zhengyuan Yang** [3]  **John Corring** [3]  **Yijuan Lu** [3]  **Jianwei Yang** [3]
**Dan Roth** [1]  **Dinei Florencio** [3]  **Cha Zhang** [3]

https://zeyofu.github.io/ReFocus/

## Abstract

Structured image understanding, such as interpreting tables and charts, requires strategically refocusing across various structures and texts within an image, forming a reasoning sequence to arrive at the final answer. However, current multimodal large language models (LLMs) lack this multihop selective attention capability. In this work, we introduce REFOCUS, a simple yet effective framework that equips multimodal LLMs with the ability to generate "visual thoughts" by performing visual editing on the input image through code, shifting and refining their visual focuses. Specifically, REFOCUS enables multimodal LLMs to generate Python codes to call tools and modify the input image, sequentially drawing boxes, highlighting sections, and masking out areas, thereby enhancing the visual reasoning process. We experiment upon a wide range of structured image understanding tasks involving tables and charts. REFOCUS largely improves performance on all tasks over GPT-4o without visual editing, yielding an average gain of 11.0% on table tasks and 6.8% on chart tasks. We present an in-depth analysis of the effects of different visual edits, and reasons why REFOCUS can improve the performance without introducing additional information. Further, we collect a 14k training set using REFOCUS, and prove that such visual chain-of-thought with intermediate information offers a better supervision than standard VQA data, reaching a 8.0% average gain over the same model trained with QA pairs and 2.6% over CoT.

---
[*]Work done during internship at Microsoft. [1]University of Pennsylvania [2]Virginia Tech [3]Microsoft. Correspondence to: Xingyu Fu <xingyuf2@seas.upenn.edu>.

*Proceedings of the $42^{nd}$ International Conference on Machine Learning*, Vancouver, Canada. PMLR 267, 2025. Copyright 2025 by the author(s).

## 1. Introduction

Structured images, such as tables and charts (Kafle et al., 2018; Chen et al., 2020; Zhu et al., 2021; Masry et al., 2022; Kim et al., 2024; Wang et al., 2024), are essential for the efficient communication of information in our daily lives (Chang et al., 2008; Groom et al., 2016). Understanding these structured images requires multiple reasoning steps – each paying **selective attention** (Duncan, 1984; Johnston & Dark, 1986; Rizzolatti et al., 1994) to focus on particular pieces of related information while ignoring less pertinent details and distractions, to facilitate multi-hop visual reasoning and arrive at the final answer. For instance, `What's total wins by Belgian riders` in Figure 1? To answer this, we might 1) identify the teams with Country being Belgium, 2) find the wins of each team, 3) locate the wins of the teams from Belgium, and 4) sum the wins together.

One major limitation of current multimodal models is their lack of selective attention and multi-hop visual reasoning ability – they limit intermediate reasoning to textual formats only. Most methods often extract the information from the image to text first, and then rely on language model's chain-of-thought (CoT) (Wei et al., 2022) reasoning to solve the real problems (Masry et al., 2023; Han et al., 2023; Liu et al., 2023c; Zhang et al., 2023; Chen et al., 2024; Surís et al., 2023; Gupta & Kembhavi, 2023), never looking back at the image again. Recent work Visual Sketchpad (Hu et al., 2024) for the first time attempts to create visual artifacts as thoughts using vision tools. However, it only works on natural images and mainly benefits from additional information brought by tools instead of from visual reasoning.

In this work, we explore an improved representation of intermediate thoughts, to boost the visual reasoning ability of multimodal large language models (LLMs) on structured image understanding. We demonstrate that simple visual editing actions generated as Python code from multimodal LLMs–such as drawing boxes, highlighting areas, or masking regions–can significantly improve model performance by directing selective attention.

To this end, we introduce REFOCUS: a framework that en-

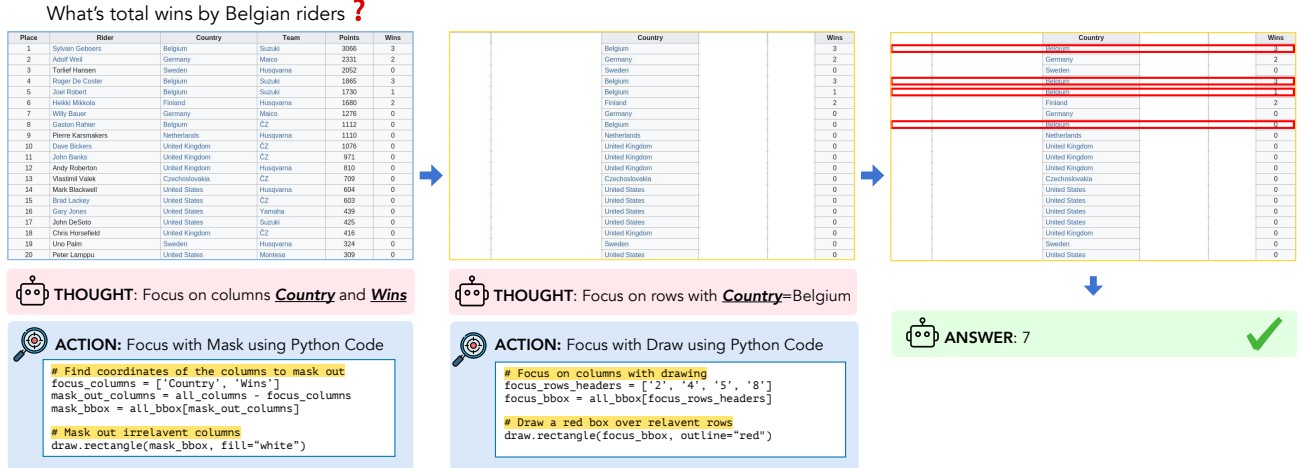

*Figure 1.* **Overview of REFOCUS.** REFOCUS performs visual chain of thought via input-image editing on an example data from TableVQA (Kim et al., 2024). Given an image and question pair, REFOCUS equips GPT-4 with editing tools (details in §3), and GPT-4 generates pseudo code if an edit action is needed. REFOCUS then executes the editing actions, and feeds GPT-4 with the new image until an answer is reached. In the above example, `mask_column` and `draw_row` are performed.

hances multimodal LLMs by integrating visual reasoning as an intermediate step. REFOCUS provides an interface that allows models to generate visual artifacts with a set of image editing tools implemented in Python code. Specifically, REFOCUS prompts the underlying multimodal LLM to program, executes the code, and produces visual artifacts that incorporate selective attention as the new input for the model. For instance, for long and complicated tabular image as in Figure 1, REFOCUS masks irrelevant columns and highlights important rows by drawing boxes around them. Similarly, to find "the average of last four countries data" in Figure 2, REFOCUS modifies the figure by removing the data of other countries. The new image therefore eliminate distractions, avoiding possible hallucinations, and become easier to solve for multimodal LLMs by refocusing.

We demonstrate the effectiveness of REFOCUS across a wide range of structured image tasks, focusing on table and chart visual question answering (VQA) sets. For tabular problems, REFOCUS enables models to selectively edit the columns and rows in the input image. REFOCUS consistently improves the baseline GPT-4o performance, yielding an average gain of **11.0%**. For chart problems, we tackle diverse types of charts including (1) horizontal bar charts, (2) vertical bar charts, and (3) complex scientific charts from arXiv papers. REFOCUS empowers multimodal LLMs to modify bars and subplots to pay selective attention, resulting in consistent improvements across different types of chart images, with the average gain over gpt-4o reaching **6.8%**. We present an in-depth analysis on REFOCUS. We first study why REFOCUS could achieve large gains, especially since it does not bring in external information as other methods (Hu et al., 2024) do, and examine how different editing techniques could affect multimodal LLMs differently.

Further, we investigate whether we can distill such refocus abilities to smaller models through supervised-finetuning (SFT), and if such visual chain-of-thought (VCoT) data could benefit models more than the standard visual question answering (VQA) data. We collect a 14k training set data including the focus area bounding boxes and reasoning processes using REFOCUS + GPT4o. Surprisingly, comprehensive experiments on Phi-3.5-vision show that our collected REFOCUS data serves a better supervision signal than standard VQA data for SFT, achieving an average gain of **8.0%** over the same model trained with the same QA data, and **2.6%** over CoT data.

To summarize, we introduce (1) REFOCUS, a simple yet effective visual reasoning framework that enhances structured image understanding through input-image editing; (2) we demonstrate that REFOCUS consistently achieves performance improvements and analyze underlying reasons behind these gains; (3) we curate a 14k training set using REFOCUS and GPT-4o, and show that model finetuned with REFOCUS data consistently outperforms the same model trained with the same set of QA or CoT data, suggesting potential pathways for more intelligent reasoning in vision language models.

## 2. Related Works

**Structured Image Understanding** Scientific chart and table figures have long been a challenging task for modern multimodal models. Previous methods mainly utilize Optical Character Recognition (OCR) (Liu et al., 2024b), by turning the image into text format and then performing textual reasoning (Liu et al., 2022; 2023a). Other methods (Masry et al., 2023; Han et al., 2023) focus on enhancing

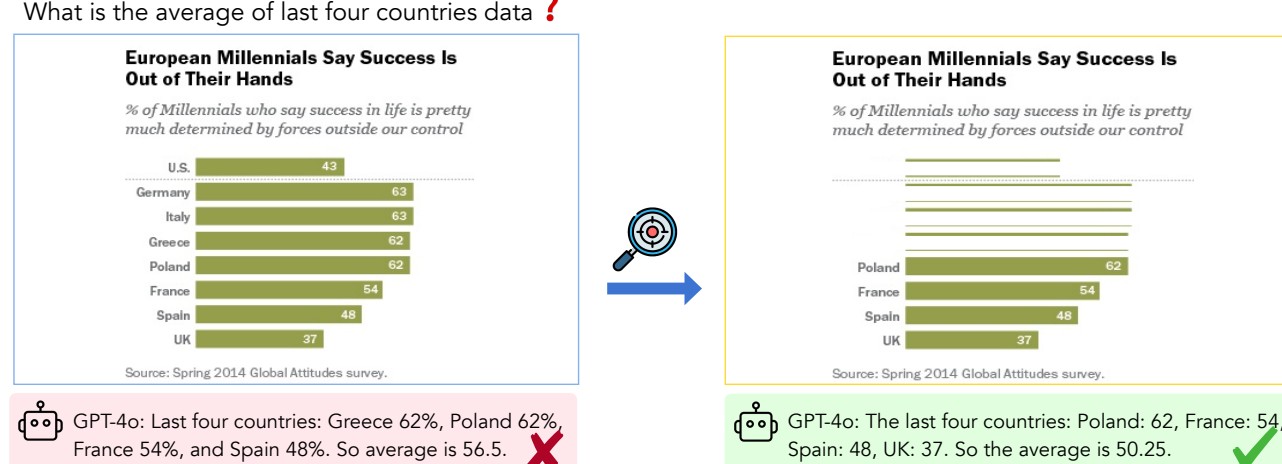

*Figure 2.* Example of how REFOCUS + GPT-4o solves previously unsolvable problem in ChartQA dataset (Masry et al., 2022) through improved **visual grounding**. Given the original horizontal bar image (left), GPT-4o grounds to the wrong bars and thus gets the wrong answer. REFOCUS eliminates such possibility through editing, guiding the model to the correct answer (right).

end-to-end VQA capabilities by training on augmented data.

**Visual Reasoning through Programming**  A significant recent trend in multimodal LLMs is using Python programs to facilitate chain-of-thought reasoning (Gupta & Kembhavi, 2023; Surís et al., 2023). However, these methods typically conduct reasoning at the text level (Fu et al., 2023; 2022; 2024a). While some methods may utilize image crops, the images themselves remain unchanged, which limits their visual reasoning capabilities. More recent work, such as Visual Sketchpad (Hu et al., 2024), advances visual reasoning by providing various computer vision tools like Segment Anything, DINO, and Depth Anything (Yang et al., 2024; Li et al., 2023; Liu et al., 2023d) to create visual artifacts. However, Visual Sketchpad mainly achieves performance gains by incorporating external expert knowledge through these tools, whereas we explore the possibility of enhancing visual understanding without additional information. Also, Visual Sketchpad cannot solve text-rich images in structured image problems, as its tools are vision-based and only address object-centric issues. With REFOCUS, LLMs can write code to call different functions for editing the input image, sequentially simplifying the original problem to make it easier for multimodal LLMs to solve. This approach enables better visual understanding and reasoning without relying on external data or expert knowledge.

**Visual Reasoning through Visualization**  It has become evident that certain types of visual prompts on the input images, such as adding a red circle around a target object, can significantly impact the performance of multimodal language models (Shtedritski et al., 2023), on certain abilities such as visual grounding (Shao et al., 2024a). The Set-of-Mark study (Yang et al., 2023) extends this investigation by demonstrating that segmenting the original image can significantly enhance the visual grounding capabilities of GPT-4v models. Meanwhile, it has been highlighted in BLINK (Fu et al., 2024b) that most open-source multimodal language models may struggle to comprehend visual prompts. Recent works, such as VIP-LLaVA and List Items One by One (Cai et al., 2024; Yan et al., 2024), have further refined these models to improve their understanding and processing of visual prompts. Most of these visual prompts are visual objects centered and do not apply to structured images. One recent work Visual Cot (Shao et al., 2024a) extracts the subarea in images where the answer corresponds to, e.g. obtaining the coordinates of answer text using OCR on chart VQA problems, or find the answer object's coordinates in the images using object detection methods. However, Visual Cot do not involve multistep reasoning and refocusing processes and only focuses on image region–answer text direct mapping. We compare our results with Visual Cot on Chart QA datasets in Section 5 and show absolute improvement.

## 3. REFOCUS

REFOCUS conducts visual editing on input images as a chain-of-thought, providing an intermediate interface for multimodal LLMs to facilitate visual reasoning and enforce the correct answer with a valid reasoning process. The entire pipeline is iterative, as illustrated in Figure 1: the multimodal LLM sees the image and question, proceeds with one step of thought, conducts visual editing using Python code, and continues with the next step of thought and editing until it arrives at the final answer. We specifically target two sets of structured image problems in this work: (1) tabular figures, and (2) chart figures. This section unfolds by introducing the details of our tasks (§3.1), the implementation of

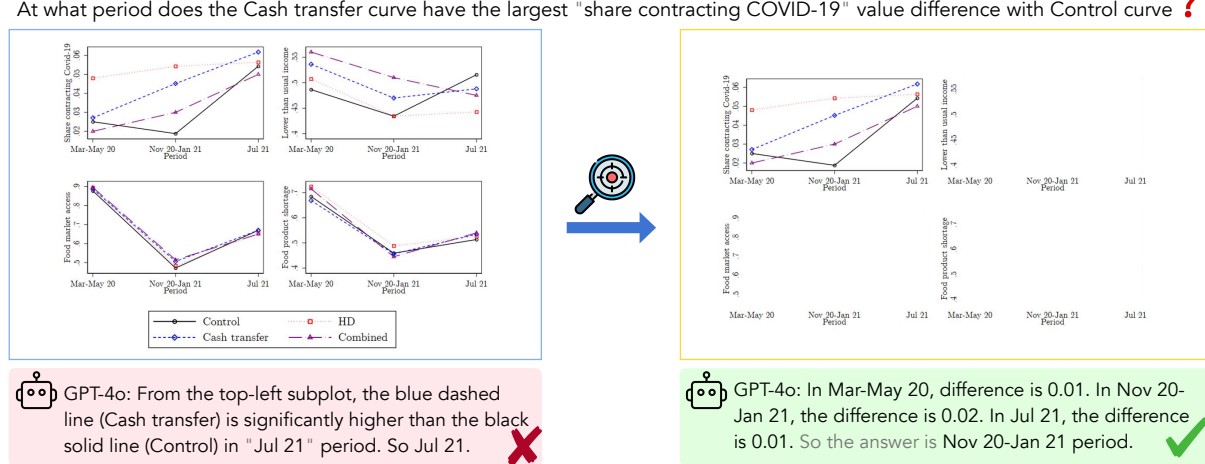

*Figure 3.* **REFOCUS equips GPT-4 with selective attention**. Above is an example of how REFOCUS + GPT-4o solves previously unsolvable problem in ChartXiv dataset (Wang et al., 2024). Specifically, REFOCUS edits upon the original image by masking out all irrelevant information – the other three subplots that could be distracting. As a result, GPT-4o is able to conduct better reasoning with the edited image, and reach the correct answer.

visual editing tools for tables (§3.2) and for charts, and how to equip multimodal LLMs with these editing tools (§3.3).

### 3.1. Structured Image Problems

**Tabular Problems**. Tabular understanding has long been a challenging task for multimodal LLMs (Liu et al., 2022; 2023a). In this paper, we use TableVQA (Kim et al., 2024) as our test-bed. We experiment on three types of table data:

**VWTQ**. VWTQ is derived from WikiTableQuestion (WTQ) (Pasupat & Liang, 2015), which provides original HTML data on Wikipedia tables along with corresponding Question Answering (QA) pairs and maintain its accuracy-based evaluation metric. (Kim et al., 2024) reproduces the original table images from WTQ by applying the stylesheet of Wikipedia to the HTML and capturing screenshots of the table images. There are a total of 750 Visual Question Answering (VQA) pairs.

**VWTQ_syn**. Considering that the table figures in VWTQ come from Wikipedia and can easily be web-crawled to gather pre-training data for multimodal LLMs, (Kim et al., 2024) generates synthetic table images using a table rendering system based on the WTQ data. This system takes HTML as input and generates tables with various styles, featuring random attributes such as background colors, border margins, and font families. There are a total of 250 Visual Question Answering (VQA) pairs in this synthetic dataset.

**VTabFact**. TabFact (Chen et al., 2019) represents a verification task that determines whether a statement is entailed or refuted given table data in text format. (Kim et al., 2024) proposes a rendering system that generates visual tables

using pseudo-HTML converted from the textual table data. There are a total of 250 Visual Question Answering (VQA) pairs in this dataset.

**Coordinate Acquisition for Tables**. To support visual editing, we need to acquire coordinates for each column and row. We accomplish this using the opencv-python[1] package with functions such as *findContours()* and *getStructuringElement()*. The heuristic behind this process is that we detect lines and boxes around text in the figure. The longest vertical line should correspond to the row length, and the longest horizontal line should correspond to the column length. By combining the longest contours in the figure, the coordinates of text boxes, and the row and column lengths, we can determine the coordinates of each column and row.

**Chart Problems**. Chart understanding is another fundamental task (Liu et al., 2022; 2023a), serving a pivotal role in real-world applications such as analyzing scientific papers or financial reports. We use the datasets CharXiv (Wang et al., 2024) and ChartQA (Masry et al., 2022) for our experiments:

**CharXiv** Multi-subplot. CharXiv is a high-quality, expert-annotated, challenging VQA benchmark that involves natural and diverse charts from arXiv papers. CharXiv includes two types of questions: descriptive questions that examine basic chart elements and reasoning questions that require synthesizing information across complex visual elements in the chart. In this paper, we specifically focus on the challenging reasoning questions that involve multiple subplots. Due to computation cost concerns, we randomly select 200

---
[1]https://pypi.org/project/opencv-python/

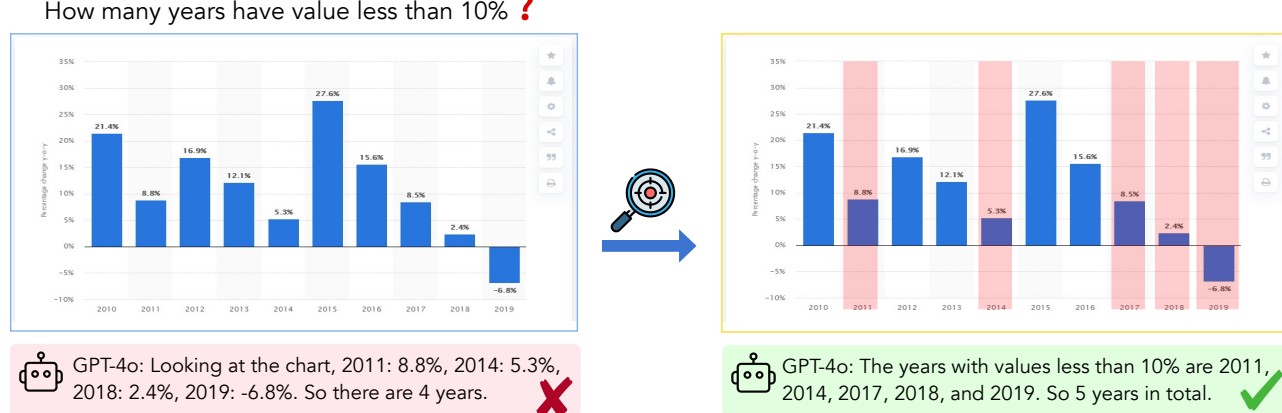

*Figure 4.* **REFOCUS unleashes better visual grounding and counting abilities for GPT-4** as in ChartQA (Masry et al., 2022) Vertical Bar problems.

VQA pairs out of the 1000 test data and use GPT-4 to select the ones with multiple subplots for experimental purposes, yielding a total of 143 VQA pairs.

**Horizontal Bar**. ChartQA (Masry et al., 2022) is one of the most popular chart-based VQA benchmarks, consisting of human-written questions focusing on logical and visual reasoning based on web-crawled diverse chart figures. For our experiments, we take the horizontal bar figures and their corresponding VQA data from the test split. This subset includes a total of 444 VQA pairs.

**Vertical Bar**. Similar to the horizontal bar figures, we also take the vertical bar figures along with their corresponding VQA data from the test split of the ChartQA (Masry et al., 2022) dataset. This subset includes a total of 382 VQA pairs.

**Coordinate Acquisition for Charts**. For CharXiv multi-subplot figures, we acquire the coordinates of each subplot using the opencv-python[1] package with functions such as *findContours()* and *getStructuringElement()*. The heuristic behind this process is that we take the top k longest contours, provide them in a prompt to LLMs, and allow the LLMs to determine which coordinates correspond to which subplot (k = 10 in our experiments). Regarding ChartQA horizontal bar figures and vertical bar figures, we utilize the provided coordinates of x-axis values and y-axis values in the original dataset. Additionally, we combine this information with coordinate details about the chart area (excluding the caption) obtained using the opencv-python[1] package.

### 3.2. Visual Editing Tools

We adopt three types of visual editing method in REFOCUS: mask out, draw box, and highlight color using Python code. In our experiments for **tabular problems**, we utilize a variety of tools as follows:

*Highlight Column*, which overlays a light red color on the columns that need to be focused on.

*Highlight Row*, which overlays a light red color on the rows that need to be focused on.

*Mask Column*, which places a white mask over the columns that do not need attention.

*Mask Row*, which places a white mask over the rows that do not need attention.

*Draw Column*, which overlays a solid red bounding box on the columns that need to be focused on.

*Draw Row*, which overlays a solid red bounding box on the rows that need to be focused on.

A pseudo code for example tool: *Highlight Column* is illustrated in Listing 1. Visual output examples can be found in Figures 1 and 7.

```python
from PIL import Image, ImageDraw

def focus_on_columns_with_highlight(image, columns_to_focus_on,
all_columns_bounding_boxes):
    """ This function is used to focus on some specific columns of the image. """

    # Draw a highlight color on the columns to focus on
    mask = image.convert('RGBA').copy()
    mask_draw = ImageDraw.Draw(mask)

    # Iterate over the columns to highlight
    for column_name in columns_to_focus_on:

        # Get the bounding box of the column
        column_bbox = all_columns_bounding_boxes[column_name]
        (x1, y1, x2, y2) = column_bbox
        mask_draw.rectangle(((x1, y1), (x2, y2)), fill=(255, 0, 0, 50))

    # Composite the overlay with the mask over the original image
    edited_image = Image.alpha_composite(image.convert('RGBA'), mask)
    return edited_image
```

*Listing 1.* Python code example for highlighting a column.

Similarly, for **chart problems**, we apply the same editing methods, but focus on different aspects: (1) subplots, particularly for CharXiv figures; (2) bars by x values, specifically for ChartQA vertical bar figures; (3) bars by y values,

| Model | Table | | | Chart | | |
|---|---|---|---|---|---|---|
| | VWTQ | VWTQ_syn | VTabFact | CharXiv | Horizontal Bar | Vertical Bar |
| *Prior Multimodal LMs* | | | | | | |
| LLaVA-NeXT-34B (Liu et al., 2024a) | 36.4 | 38.0 | 71.2 | 18.9 | 23.4 | 12.6 |
| Phi 3 vision (Abdin et al., 2024) | 44.7 | 53.2 | 74.4 | 16.2 | 60.8 | 66.5 |
| Gemini-Pro 1.5 (Team et al., 2023) | 38.5 | 43.2 | 75.6 | 38.3 | 57.2 | 66.0 |
| VisProg (Gupta & Kembhavi, 2023) | 53.2 | 62.0 | 76.4 | 46.8 | 69.8 | 68.6 |
| *Latest multimodal LLMs + REFOCUS* | | | | | | |
| gpt-4o-2024-05-13 (OpenAI, 2023) | 66.5 | 73.2 | 89.6 | 49.0 | 78.2 | 76.2 |
| + REFOCUS | 76.9 | 79.6 | 89.6 | **57.3** | **85.4** | 81.0 |
| | +10.4 | +3.4 | +0.0 | +8.3 | +7.2 | +4.8 |
| gpt-4o-2024-08-06 (OpenAI, 2023) | 66.4 | 70.4 | 90.0 | 48.9 | 75.2 | 74.9 |
| + REFOCUS | **77.2** | **82.8** | **90.8** | 46.2 | 82.0 | **81.2** |
| | +9.8 | +12.4 | +0.8 | -2.7 | +5.0 | +4.2 |

*Table 1.* REFOCUS yields consistent performance gains across all tasks and outperforms all baselines. Notice that the GPT baselines here are also in a conversational format but without editing abilities. For fair comparison, we modify the original Visprog (Gupta & Kembhavi, 2023) framework by replacing the LM and VQA components with the latest GPT-4o model.

specifically for ChartQA horizontal bar figures. Each of these aspects is combined with the three editing methods: masking, drawing, and highlighting. Detailed visual output examples can be found in Figures 2 to 4, and a complete list of tools is available in Appendix B.

### 3.3. Equip LLMs with Visual Editing Tools

We provide the function names of each visual editing tool in the multimodal LLM's prompts and ask it to generate pseudocode using these provided function names. The actual editing functions will then be executed when the multimodal LLM decides to perform a visual editing, and the modified images are returned to the model as the new input. More prompting details can be found in Appendix C.

## 4. Experiments and Analyses

We demonstrate that REFOCUS significantly improves multimodal LLMs' performance on structured images, while making the visual reasoning processes interpretable. In this section, we introduce the baseline models and experiment settings (§4.1), present comprehensive experiment results (§4.2), and show an in-depth analyses investing why REFOCUS is so effective despite its simplicity(§4.3).

### 4.1. Baselines and Setups

We apply REFOCUS on the state-of-the-art multimodal LLM: GPT-4o (OpenAI, 2023), especially the gpt-4o-2024-05-13 and gpt-4o-2024-08-06 checkpoints to show its efficacy. We compare with the default GPT-4o for both checkpoints using chain-of-thought prompting for fair comparison. We also include several other powerful multimodal LLMs

to compare with: LLaVA-NeXT (Liu et al., 2024a), Phi 3 Vision[2](Abdin et al., 2024), and Gemini Pro 1.5 (Team et al., 2023). In addition, we also compare our approach with the visual programming method VisProg (Gupta & Kembhavi, 2023), which uses Python code to generate intermediate reasoning with integrated vision modules. We modify the original VisProg framework by replacing the LLM and VQA components with the latest gpt-4o-2024-08-06 model for fair comparison. We do not include Visual Sketchpad (Hu et al., 2024) since none of its tools can apply to structured image problems, leaving it the same as vanilla GPT-4o. More implementation details are in Appendix B.

### 4.2. Results

As shown in Table 1, the mean accuracy of REFOCUS combined with GPT-4o consistently surpasses all baseline models, including the vanilla conversational GPT-4o model without editing abilities. REFOCUS is particularly effective on VWTQ, VWTQ_syn, Horizontal Bar, and Vertical Bar tasks, achieving a consistent 5-10% accuracy improvement over GPT-4o. To examine the quality of visual editings, as illustrated in Table 2, we further test LLaVA-NeXT (7B, 13B, 34) and Phi-3-vision models using the REFOCUS + GPT-4o edited images. Interestingly, while none of the tested models are fine-tuned on images with visual prompts, we still observe a consistent improvement on VWTQ, VWTQ_syn, CharXiv, and Vertical Bar tasks. In Table 5, we compare the performance of REFOCUS versus vanilla GPT-4o with different inputs (i.e., gold table / chart text, figure, or gold text + figure). Surprisingly, GPT-4o with REFOCUS often outperforms GPT-4o with gold text and figure inputs by a

---

[2]We used the checkpoint at `https://huggingface.co/microsoft/Phi-3-vision-128k-instruct`.

| Model | Table | | | Chart | | |
|---|---|---|---|---|---|---|
| | VWTQ | VWTQ_syn | VTabFact | CharXiv | Horizontal Bar | Vertical Bar |
| *Multimodal LLMs with original visual inputs* | | | | | | |
| LLaVA-NeXT-7B (Liu et al., 2024a) | 21.7 | 24.0 | 56.8 | 16.1 | 8.1 | 7.3 |
| LLaVA-NeXT-13B (Liu et al., 2024a) | 25.6 | 30.4 | 62.4 | 18.9 | 13.5 | 13.1 |
| LLaVA-NeXT-34B (Liu et al., 2024a) | 36.4 | 38.0 | 71.2 | 18.9 | 23.4 | 12.6 |
| Phi 3 vision (Abdin et al., 2024) | 44.7 | 53.2 | **74.4** | 16.2 | **60.8** | **66.5** |
| *Multimodal LLMs with REFOCUS edited visual inputs* | | | | | | |
| LLaVA-NeXT-7B + Oracle REFOCUS | 25.7 | 26.8 | 55.2 | 15.4 | 6.8 | 8.4 |
| LLaVA-NeXT-13B + Oracle REFOCUS | 30.3 | 31.6 | 61.2 | 17.5 | 15.5 | 13.1 |
| LLaVA-NeXT-34B + Oracle REFOCUS | 39.1 | 41.2 | 68.0 | **21.0** | 26.1 | 15.2 |
| Phi 3 vision + Oracle REFOCUS | **48.3** | **56.4** | 72.8 | 17.6 | 60.4 | **66.5** |

*Table 2.* Open-source models' performance upon the original visual input versus GPT-4o + REFOCUS edited images as visual input (referred by + oracle REFOCUS). Notice that + oracle REFOCUS uses the visual artifact generated in the last action of GPT-4o + REFOCUS as inputs.

significant margin on most tasks. This demonstrates that REFOCUS can effectively enhance GPT-4o's structured image understanding capability, making it perform as if it had access to the gold text input.

### 4.3. Analyses

**Why can REFOCUS improve the performance?** This question is interesting because REFOCUS does not introduce any additional information as other methods (Yang et al., 2023; Hu et al., 2024) do. Still, it achieves an impressive performance gain upon the already powerful GPT-4o models, using a simple strategy. Our observations on the intermediate outputs suggest that this might be because REFOCUS improves GPT-4o's visual grounding and OCR abilities through selective attention and eliminates hallucinations. As shown in Figure 2, for the original image on the left, GPT-4o mistakenly grounds to Greece and misses the UK, resulting in the wrong answer of 56.5. REFOCUS decomposes the steps of visual grounding and optical character recognition (OCR). In the edited image on the right, only the last four countries are shown, and GPT-4o successfully provides the correct answer with proper reasoning. Similarly, as shown in Figure 7, when GPT-4o tries to recognize characters in the original image, it makes a spelling mistake, identifying "Partenavia" incorrectly. However, with the highlighted columns, GPT-4o can correctly recognize the characters as "Partenvia."

**Which editing method works the best?** Since we provide three types of editing methods: draw box, highlight color, and mask out, it raises a natural question about which method works best for the models. We conducted experiments on VWTQ and VWTQ_syn sets and show the performance differences in Table 3. In short, all tools are similar.

| Dataset | Original | Mask out | Draw Box | Highlight |
|---|---|---|---|---|
| VWTQ | 66.4 | 77.2 | **77.6** | 74.8 |
| VWTQ_syn | 70.4 | **82.4** | 78.8 | 80.8 |

*Table 3.* Analysis on how different editing tool can affect model performance. We control the tool type provided by REFOCUS and experiment on the VWTQ and VWTQ_syn datasets.

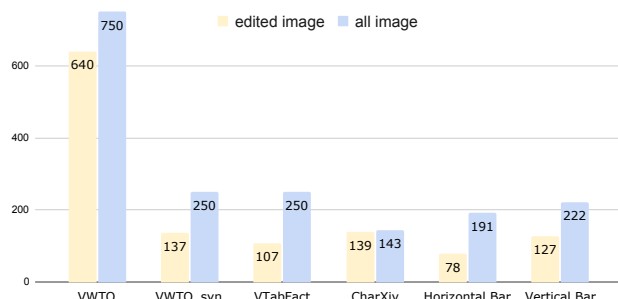

*Figure 5.* Statistics of how often visual editing are performed.

**How frequent is visual editing performed?** Since the visual editing steps are conducted through LLM generated python code, it is possible that REFOCUS does not conduct any editing for some instances. To answer how often the underlying multimodal LLM, GPT-4o in our case, decides to proceed with a visual edit, we calculated the number of images that GPT-4o decides to edit and report the numbers in Figure 5. GPT-4o clearly shows a preference to edit > 85% VWTQ and CharXiv images, while the portion of edited images on the other datasets remains around 40-55%.

## 5. Finetune with REFOCUS data

While REFOCUS shows its efficacy, it is still limited to a visual prompting method. Could the visual reasoning processes of REFOCUS be distilled into smaller Multimodal LLMs and provide additional or even better supervision for

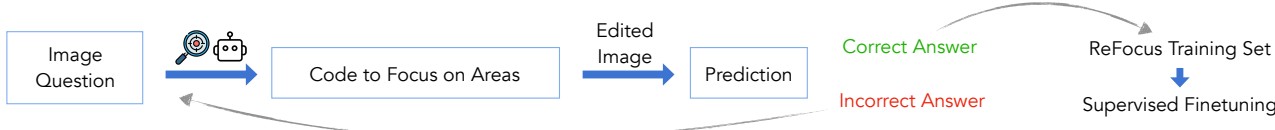

*Figure 6.* **Training set collection using REFOCUS on ChartQA dataset.**

models to learn? What would be the difference between a model fine-tuned on the de facto image question answer pairs, versus one fine-tuned on CoT data, or on REFOCUS visual reasoning data?

**Finetune Data Collection** To answer this question, we further collect a 14k training set using REFOCUS and GPT-4o upon the ChartQA (Masry et al., 2022) training data. As shown in Figure 6, we apply the pipeline on 15,059 ChartQA training data. If the prediction is correct, we keep the whole process and add the generated textual chain-of-thought (CoT) reasoning, python code for editing, and the area to focus on (in bounding box format) to our data. If the prediction is incorrect, we give it one more try and hint it with the gold answer. If it still gets to a wrong answer, we discard this example since it's most likely a data noise from our observation. We end up with 14,344 training data, among which 12,819 has editing processes. More dataset details are in Appendix D.

**Finetune Setups** To test the effectiveness of visual chain-of-thought for multimodal LLMs, we adopt the standard supervised-Finetuning (SFT) strategies. We use Phi-3.5-vision[3] (Abdin et al., 2024) as our base model, and compare its performance when finetuned with REFOCUS data, versus with same set of data but only with QA pairs. For both finetuning, we keep all settings the same and search through the same set of hyper-parameters for learning rate and epoch number, and report the best performance. During training time, the input is given in the format of '<image> <question> <thought1> <ReFocus bounding box> <thought2> <answer>', where <Thought1> is transformed through GPT-4o returned thought on what areas to refocus on, and <Thought2> is GPT-4o output when predicting based on the edited images. In comparison, the standard QA training input is '<image> <question> <answer>'. During inference time, we examine two types of prompt settings: '<image> <question> Answer:' as the QA prompting, and '<image> <question> Thought:' as the Visual CoT prompting. More details can be found in Appendix D.

**Supervised-Finetuning Results** We report our results as in

---

[3] https://github.com/microsoft/Phi-3CookBook/blob/main/md/04.Fine-tuning/FineTuning_Vision.md

| | Horizontal Bar | Vertical Bar | Avg. |
|---|---|---|---|
| *QA Prompting* | | | |
| Phi-3.5-vision | 60.1 | 63.1 | 61.5 |
| SFT w/ QA Data | 60.1 | 65.5 | 62.6 |
| *Visual CoT Prompting* | | | |
| Visual CoT 7b-336 | 27.1 | 14.5 | 20.8 |
| Phi-3.5-vision | 69.4 | 66.8 | 68.2 |
| SFT w/ QA Data | 60.6 | 66.8 | 63.4 |
| SFT w/ REFOCUS CoT | 67.1 | 70.7 | 68.8 |
| SFT w/ REFOCUS VCoT | **71.0** | **72.2** | **71.4** |

*Table 4.* **SFT accuracy results.** The difference between REFOCUS VCoT data and CoT data is that VCoT contains refocus area bounding box coordinates whereas CoT does not. All trainings select the best performing hyper-parameters on the same set of training data.

Table 4 with multiple setups – the REFOCUS data is Visual CoT (VCoT) data, and we compare with the standard QA data. Another setup is finetuning with CoT data, which is VCoT data without refocus area bounding box information. The Phi-3.5-Vision model finetuned with REFOCUS VCoT data outperforms the base model by 3.2% in accuracy; over the same model finetuned with same set of QA data by 8.0%; and over the model finetuned with CoT data by 2.6%. Notably, our results largely exceed Visual Cot (Shao et al., 2024b) which trains upon LLaVa-1.5 (Liu et al., 2023b) with answer coordinate data by 50.6%. These results suggest that implicit visual reasoning data such as REFOCUS VCoT can consistently provide better supervision signals than standard QA pairs and CoT data for structured image understanding tasks. Our approach has the potential to provide rich and meaningful supervision data to enhance multimodal LLMs in general. More SFT details, result analyses, and image editing outputs can be found in Appendices D and E.

## 6. Conclusion

This work introduces REFOCUS, a simple yet effective framework that boosts multimodal large language models (LLMs) to understand structured images better, by incorporating visual editing on the input image through Python code. Our approach significantly boosts performance on table and chart tasks. Further, we present a 14k visual chain-of-thought training data built using REFOCUS, which demonstrates superior supervision over standard VQA data.

## Acknowledgements

We would like to thank Weijian Xu, Guoxin Wang, Yushi Hu and Weijia Shi for their helpful and insightful discussions. Xingyu Fu and Dan Roth were funded by ONR Contract N00014-23-1-2364, ONR Contract N00014-23-1-2417, and NSF grant IIS-2212433.

## Impact Statement

This paper presents work whose goal is to advance the field of Machine Learning. There are many potential societal consequences of our work, none which we feel must be specifically highlighted here.

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

## A. Overview

In the supplemental materials, Appendix B contains additional details for REFOCUS, including editing tools(§B.3), and experiment configurations such as baseline model setups (§B.4). Appendix D discusses supervised fine-tuning details including experiment setups(§D.2), dataset statistics(§D.1), and result analyses(§D.3). Appendix C demonstrates prompts in REFOCUS. Appendix E shows qualitative SFT output examples.

## B. REFOCUS Prompting Details

### B.1. Additional Example

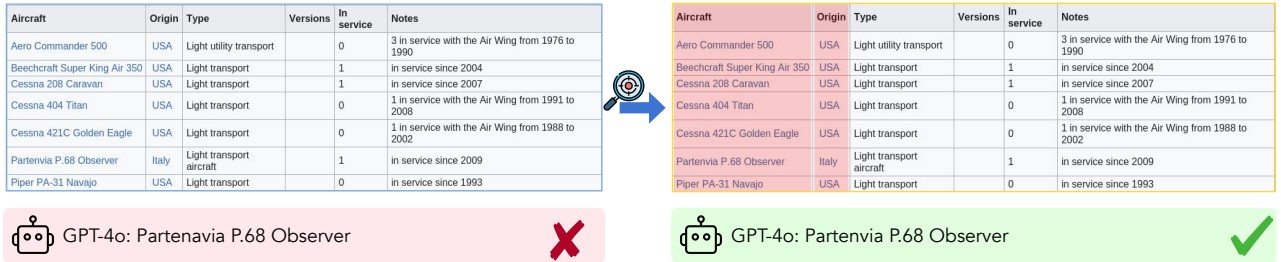

*Figure 7.* **REFOCUS unleashes better OCR for GPT-4.** In this example from TableVQA (Kim et al., 2024), REFOCUS + GPT-4 conducts the edit action `highlight_column`. With this simple action, GPT-4 can focus more on the important subarea, and recognize the characters better.

### B.2. Additional Analysis

| Model | Table | | | Chart | | |
| --- | --- | --- | --- | --- | --- | --- |
| | VWTQ | VWTQ_syn | VTabFact | CharXiv | Horizontal Bar | Vertical Bar |
| *GPT-4o Model* | | | | | | |
| Text input | 68.1 | 69.6 | 80.0 | \ | 66.2 | 74.9 |
| Figure Input | 61.0 | 68.4 | 88.4 | 47.9 | 75.2 | 74.9 |
| Text + Figure Input | 67.5 | 72.8 | 91.6 | \ | 75.7 | 81.7 |
| *GPT-4o + REFOCUS* | | | | | | |
| Figure Input | 77.2 | 82.8 | 90.8 | 57.3 | 85.4 | 81.2 |
| | +16.2 | +14.4 | +2.4 | +10.6 | +9.7 | +0.5 |

*Table 5.* **REFOCUS empowers GPT-4o to achieve the performance as if given gold text input.** The text input are mainly csv tables, as detailed in Section 3.1. Notice that Table 1 reports the conversational performance without visual editing, whereas the performance discussed here is based on direct question answering, leading to minor differences.

### B.3. Visual Editing Tools for Charts

As introduced in Section 3.2, we adopt three types of visual editing method in REFOCUS: mask out, draw box, and highlight color using Python code. In our experiments for **chart problems**, we deploy a list of tools as follows:

*Highlight Bar at X*, which overlays a light red color over a bar in a vertical bar figure that needs to be focused. It operates based on the x-value of the chosen bar, as in Figure 4.

*Highlight Bar at Y*, which overlays a light red color over a bar in a horizontal bar figure that needs to be focused. It operates based on the y-value of the chosen bar, as in Figure 2.

*Mask Bar at X*, which places a white mask over bars in a vertical bar figure that do not need attention.

*Mask Bar at Y*, which places a white mask over bars in a horizontal bar figure that do not need attention.

*Draw Bar at X*, which overlays a solid red bounding box over bars in a vertical bar figure that need to be focused on.

*Draw Bar at Y*, which overlays a solid red bounding box over bars in a horizontal bar figure that need to be focused on.

### B.4. Experiment Details

For all the experiments, the temperature is set to 0. All evaluations are done using GPT-4, where it's given a prediction and the gold answer, and decides whether the prediction is correct or not. For the Section 4 experiments, we use 4 NVIDIA Quadro RTX 8000 GPUs for the inference of open-source multimodal LLMs, including LLaVA-NeXT-7B, LLaVA-NeXT-13B, LLaVA-NeXT-34B, and Phi 3 vision (4B). These experiments cost around 40 GPU hours.

# C. Prompts

We show REFOCUS prompting details for table problems and chart problems with running examples randomly selected from VWTQ and Horizontal Bar datasets as follows.

**Prompts for Table Problems:**

---

**SYSTEM PROMPT**
You are a helpful multimodal AI assistant. [MORE INSTRUCTIONS ...]
For each turn, you should first do a "THOUGHT", based on the images and text you see. If you think you get the answer to the intial user request, you can reply with "ANSWER: <your answer>" and ends with "TERMINATE".

---

**Initial Prompt + Request**

---

```python
1   Here are some tools that can help you. All are python codes. They are in tools .py and will be imported for you.
2   You will be given a table figure : image_1 and a question .
3   Notice that you, as an AI assistant , are not good at answering questions when there are too many unnecessary and irrelevant information . You should determine which are the relevant
      columns to the question , and specify them in a python list . You should use the given column headers.
4   You should also determine which are the relevant rows to the question , and specify them in a python list . You should use the given row headers .
5   You could select the tools to focus on some columns / rows, or mask out some columns / rows. Use whichever tool you think is more appropriate . Below are the tools in tools .py:
6   ```python
7   def focus_on_columns_with_highlight(image, columns_to_focus_on, all_columns_bounding_boxes):
8       """
9       This function is useful when you want to focus on some specific columns of the image.
10      It does this by adding light transparent red highlight to the columns that need to be focused on.
11      For example, you can focus on the columns in a table that are relevant to your analysis .
12      Return the drawed image.
13
14      Args:
15          image (PIL.Image.Image): the input image
16          columns_to_mask (List[ str ]) : a list of column names to focus on.
17          all_columns_bounding_boxes (Dict[Dict ]]) : a dictionary of bounding boxes for all columns in the image. key is column name and value is the bounding box of that column. Each
      bounding box is in the format {'x1': x1, 'y1': y1, 'x2': x2, 'y2': y2}.
18
19      Returns :
20          image_with_focused_columns (PIL.Image.Image): the image with specified columns focused on
21
22      Example:
23          image = Image.open("sample_img.jpg")
24          image_with_focused_columns = focus_on_columns_with_highlight(image, ["Year", "Name"], {"Year": {'x1': 0.1, 'y1': 0.1, 'x2': 0.3, 'y2': 0.9}, "Team": {'x1': 0.4, 'y1': 0.1,
      'x2': 0.6, 'y2': 0.9}, "Name": {'x1': 0.7, 'y1': 0.1, 'x2': 0.9, 'y2': 0.9}})
25          display (image_with_focused_columns)
26      """
27
28  def focus_on_rows_with_highlight (image, rows_to_focus_on, all_rows_bounding_boxes):
29      """
30      This function is useful when you want to focus on some specific rows of the image.
31      It does this by adding light transparent red highlight to the rows that need to be focused on.
32      For example, you can focus on the rows in a table that are relevant to your analysis .
33      Return the drawed image.
34
35      Args:
36          image (PIL.Image.Image): the input image
37          rows_to_focus_on (List[ str ]) : a list of row headers to focus on.
38          all_rows_bounding_boxes (Dict[Dict ]): a dictionary of bounding boxes for all rows in the image. key is row header and value is the bounding box of that row. Each bounding
      box is in the format {'x1': x1, 'y1': y1, 'x2': x2, 'y2': y2}.
39
40      Returns :
41          image_with_focused_rows (PIL.Image.Image): the image with specified rows focused on
42
43      Example:
44          image = Image.open("sample_img.jpg")
45          image_with_focused_rows = focus_on_rows_with_highlight(image, ["1972"], ["Year": {'x1': 0.1, 'y1': 0.1, 'x2': 0.9, 'y2': 0.15}, "1969": {'x1': 0.1, 'y1': 0.2, 'x2': 0.9, '
      y2': 0.5}, "1972": {'x1': 0.1, 'y1': 0.6, 'x2': 0.9, 'y2': 0.9}])
46          display (image_with_focused_rows)
47      """
48
49  def focus_on_columns_with_mask(image, columns_to_focus_on, all_columns_bounding_boxes):
50      """
51      This function is useful when you want to focus on some specific columns of the image.
52      It does this by masking out the columns that are not needed.
53      For example, you can focus on the columns in a table that are relevant to your analysis and ignore the rest .
54      Return the masked image.
55
56      Args:
57          image (PIL.Image.Image): the input image
58          columns_to_mask (List[ str ]) : a list of column names to focus on.
59          all_columns_bounding_boxes (Dict[Dict ]]) : a dictionary of bounding boxes for all columns in the image. key is column name and value is the bounding box of that column. Each
      bounding box is in the format {'x1': x1, 'y1': y1, 'x2': x2, 'y2': y2}.
60
61      Returns :
62          image_with_focused_columns (PIL.Image.Image): the image with specified columns focused on
63
64      Example:
65          image = Image.open("sample_img.jpg")
66          image_with_focused_columns = focus_on_columns(image, ["Year", "Name"], {"Year": {'x1': 0.1, 'y1': 0.1, 'x2': 0.3, 'y2': 0.9}, "Team": {'x1': 0.4, 'y1': 0.1, 'x2': 0.6, 'y2
      ': 0.9}, "Name": {'x1': 0.7, 'y1': 0.1, 'x2': 0.9, 'y2': 0.9}})
67          display (image_with_focused_columns)
68      """
```

```python
1
2  def focus_on_rows_with_mask(image, rows_to_focus_on, all_rows_bounding_boxes):
3      """
4      This function is useful when you want to focus on some specific rows of the image.
5      It does this by masking out the rows that are not needed.
6      For example, you can focus on the rows in a table that are relevant to your analysis and ignore the rest.
7      Return the masked image.
8
9      Args:
10         image (PIL.Image.Image): the input image
11         rows_to_focus_on (List[str]): a list of row headers to focus on.
12         all_rows_bounding_boxes (Dict[Dict]): a dictionary of bounding boxes for all rows in the image. key is row header and value is the bounding box of that row. Each bounding
    box is in the format {'x1': x1, 'y1': y1, 'x2': x2, 'y2': y2}.
13
14     Returns:
15         image_with_focused_rows (PIL.Image.Image): the image with specified rows focused on
16
17     Example:
18         image = Image.open("sample_img.jpg")
19         image_with_focused_rows = focus_on_rows(image, ["1972"], ["Year": {'x1': 0.1, 'y1': 0.1, 'x2': 0.9, 'y2': 0.15}, "1969": {'x1': 0.1, 'y1': 0.2, 'x2': 0.9, 'y2': 0.5},
    "1972": {'x1': 0.1, 'y1': 0.6, 'x2': 0.9, 'y2': 0.9}])
20         display(image_with_focused_rows)
21     """
22
23 def focus_on_columns_with_draw(image, columns_to_focus_on, all_columns_bounding_boxes):
24     """
25     This function is useful when you want to focus on some specific columns of the image.
26     It does this by drawing a red box around the columns that need to be focused on.
27     For example, you can focus on the columns in a table that are relevant to your analysis.
28     Return the drawed image.
29
30     Args:
31         image (PIL.Image.Image): the input image
32         columns_to_mask (List[str]): a list of column names to focus on.
33         all_columns_bounding_boxes (Dict[Dict]]): a dictionary of bounding boxes for all columns in the image. key is column name and value is the bounding box of that column. Each
    bounding box is in the format {'x1': x1, 'y1': y1, 'x2': x2, 'y2': y2}.
34
35     Returns:
36         image_with_focused_columns (PIL.Image.Image): the image with specified columns focused on
37
38     Example:
39         image = Image.open("sample_img.jpg")
40         image_with_focused_columns = focus_on_columns(image, ["Year", "Name"], {"Year": {'x1': 0.1, 'y1': 0.1, 'x2': 0.3, 'y2': 0.9}, "Team": {'x1': 0.4, 'y1': 0.1, 'x2': 0.6, 'y2
    ': 0.9}, "Name": {'x1': 0.7, 'y1': 0.1, 'x2': 0.9, 'y2': 0.9}})
41         display(image_with_focused_columns)
42     """
43
44 def focus_on_rows_with_draw(image, rows_to_focus_on, all_rows_bounding_boxes):
45     """
46     This function is useful when you want to focus on some specific rows of the image.
47     It does this by drawing a red box around the rows that need to be focused on.
48     For example, you can focus on the rows in a table that are relevant to your analysis.
49     Return the drawed image.
50
51     Args:
52         image (PIL.Image.Image): the input image
53         rows_to_focus_on (List[str]): a list of row headers to focus on.
54         all_rows_bounding_boxes (Dict[Dict]): a dictionary of bounding boxes for all rows in the image. key is row header and value is the bounding box of that row. Each bounding
    box is in the format {'x1': x1, 'y1': y1, 'x2': x2, 'y2': y2}.
55
56     Returns:
57         image_with_focused_rows (PIL.Image.Image): the image with specified rows focused on
58
59     Example:
60         image = Image.open("sample_img.jpg")
61         image_with_focused_rows = focus_on_columns_with_highlight(image, ["1972"], {"Year": {'x1': 0.1, 'y1': 0.1, 'x2': 0.9, 'y2': 0.15}, "1969": {'x1': 0.1, 'y1': 0.2, 'x2': 0.9,
    'y2': 0.5}, "1972": {'x1': 0.1, 'y1': 0.6, 'x2': 0.9, 'y2': 0.9}])
62         display(image_with_focused_rows)
63     """
```

64  # GOAL #: Based on the above tools, I want you to reason about how to solve the # USER REQUEST # and generate the actions step by step (each action is a python jupyter notebook
    code block) to solve the request.
65  You may need to use the tools above to process the images and make decisions based on the visual outputs of the previous code blocks.
66  Your visual ability is not perfect, so you should use these tools to assist you in reasoning about the images.
67  The jupyter notebook has already executed the following code to import the necessary packages:

```python
from PIL import Image
from IPython.display import display
from tools import *
```

75  # REQUIREMENTS #:
76  1. The generated actions can resolve the given user request # USER REQUEST # perfectly. The user request is reasonable and can be solved. Try your best to solve the request.
77  2. The arguments of a tool must be the same format specified in # TOOL LIST #;
78  3. If you think you got the answer, use ANSWER: <your answer> Please extract the final answer in FINAL ANSWER: <final answer> and ends with TERMINATE.
79  4. All images in the initial user request are stored in PIL Image objects named image_1, image_2, ..., image_n. You can use these images in your code blocks. Use display() function
    to show the image in the notebook for you too see.
80  5. Use as few tools as possible. Only use the tools for the use cases written in the tool description. You can use multiple tools in a single action.
81  6. If you have multiple answers, please separate them with ‖ marks. For example, if the answer is 'Alice' and 'Bob', you should write 'Alice ‖ Bob'.
82  7. When you focus on columns in the image, most like you need to look at multiple columns instead of a single one.
83  8. If you do not think you have enough information to answer the question on the images returned by the tools, you should directly answer the question based on the original image.
84  Below are some examples of how to use the tools to solve the user requests. You can refer to them for help. You can also refer to the tool descriptions for more information.

```
1  # [four in−context examples here]
2
3  # USER REQUEST #:
```

| Golfer | Country | Wins | Match Play | Championship | Invitational | Champions |
|---|---|---|---|---|---|---|
| Woods, TigerTiger Woods | United States | 18 | **3:** 2003, 2004, 2008 | **7:** 1999, 2002, 2003, 2005, 2006, 2007, 2013 | **8:** 1999, 2000, 2001, 2005, 2006, 2007, 2009, 2013 | — |
| Ogilvy, GeoffGeoff Ogilvy | Australia | 3 | **2:** 2006, 2009 | **1:** 2008 | — | — |
| Clarke, DarrenDarren Clarke | Northern Ireland | 2 | **1:** 2000 | — | **1:** 2003 | — |
| Els, ErnieErnie Els | South Africa | 2 | — | **2:** 2004, 2010 | — | — |
| Mahan, HunterHunter Mahan | United States | 2 | **1:** 2012 | — | **1:** 2010 | — |
| Mickelson, PhilPhil Mickelson | United States | 2 | — | **1:** 2009 | — | **1:** 2009 |
| Poulter, IanIan Poulter | England | 2 | **1:** 2010 | — | — | **1:** 2012 |

```
1  Tell me the number of wins els had.
2
3  # USER Bounding Box Info: columns_bbox, where keys are column headers and values are column bounding boxes. rows_bbox, where keys row headers and values are row bounding boxes. The
   columns in the image are: ['Golfer', 'Country', 'Wins', 'Match Play', 'Championship', 'Invitational', 'Champions']. The rows in the image start with: ['Golfer', 'Woods, TigerTiger
   Woods', 'Ogilvy, GeoffGeoff Ogilvy', 'Clarke, DarrenDarren Clarke', 'Els, ErnieErnie Els', 'Mahan, HunterHunter Mahan', 'Mickelson, PhilPhil Mickelson', 'Poulter, IanIan Poulter'].
4
5  # USER IMAGE stored in image_1, as PIL image.
6  Now please generate only THOUGHT 0 and ACTION 0 in RESULT. If no action needed, also reply with ANSWER: <your answer> Please extract the final answer in FINAL ANSWER: <final
   answer> and ends with TERMINATE:
7
8  # RESULT #:
```

### Thought + Action 0

```
1  THOUGHT 0: To find the number of wins Els had, I need to focus on the 'Wins' column and the row for 'Els, ErnieErnie Els'.
2
3  ACTION 0:
4  ```python
5  image_with_focused_columns = focus_on_columns_with_draw(image_1, ["Wins"], columns_bbox)
6  image_with_focused_rows = focus_on_rows_with_draw(image_with_focused_columns, ["Els, ErnieErnie Els"], rows_bbox)
7  display(image_with_focused_rows)
8  ```
```

### Observation 0

Execution success. The output is as follows.

| Golfer | Country | Wins | Match Play | Championship | Invitational | Champions |
|---|---|---|---|---|---|---|
| Woods, TigerTiger Woods | United States | 18 | **3:** 2003, 2004, 2008 | **7:** 1999, 2002, 2003, 2005, 2006, 2007, 2013 | **8:** 1999, 2000, 2001, 2005, 2006, 2007, 2009, 2013 | — |
| Ogilvy, GeoffGeoff Ogilvy | Australia | 3 | **2:** 2006, 2009 | **1:** 2008 | — | — |
| Clarke, DarrenDarren Clarke | Northern Ireland | 2 | **1:** 2000 | — | **1:** 2003 | — |
| Els, ErnieErnie Els | South Africa | 2 | — | **2:** 2004, 2010 | — | — |
| Mahan, HunterHunter Mahan | United States | 2 | **1:** 2012 | — | **1:** 2010 | — |
| Mickelson, PhilPhil Mickelson | United States | 2 | — | **1:** 2009 | — | **1:** 2009 |
| Poulter, IanIan Poulter | England | 2 | **1:** 2010 | — | — | **1:** 2012 |

Answer the question tell me the number of wins els had. You can turn the table image into text and answer with step of thinking.
Reply with ANSWER: <your answer>
Please extract the final answer in FINAL ANSWER: <final answer> and ends with TERMINATE.

### Thought + Action 1

ANSWER: The number of wins Els had is 2. FINAL ANSWER: 2. TERMINATE

**Prompts for Chart Problems:**

---

**SYSTEM PROMPT – CHARTS**
You are a helpful multimodal AI assistant. [MORE INSTRUCTIONS ...]
For each turn, you should first do a "THOUGHT", based on the images and text you see. If you think you get the answer to the intial user request, you can reply with "ANSWER: <your answer>" and ends with "TERMINATE".

---

**Initial Prompt + Request**

```
1   Here are some tools that can help you. All are python codes. They are in tools.py and will be imported for you. You will be given a chart figure: image_1 and a question.
2   Notice that you, as an AI assistant, are not good at answering questions when there are too many unnecessary and irrelevant information.
3   If you are dealing with a vertical bar chart figure, you should determine which are the relevant x values to the question, and specify them in a python list. You should use the
    given x value names.
4   If you are dealing with a horizontal bar chart figure, you should also determine which are the relevant y values to the question, and specify them in a python list. You should use
    the given y value names.
5   Below are the tools in tools.py:
6   ```python
7   def focus_on_x_values_with_mask(image, x_values_to_focus_on, all_x_values_bounding_boxes):
8       """
9       This function is useful when you want to focus on some specific x values in the image.
10      It does this by masking out the x values that are not needed.
11      This function is especially useful for vertical bar charts.
12      For example, you can focus on the x values in a chart that are relevant to your analysis and ignore the rest.
13      Return the masked image.
14
15      Args:
16          image (PIL.Image.Image): the input image
17          x_values_to_focus_on (List[str]): a list of x values to focus on.
18          all_x_values_bounding_boxes (Dict[Dict]): a dictionary of bounding boxes for all x values in the image. key is x value and value is the bounding box of that x value. Each
    bounding box is in the format {'x1': x1, 'y1': y1, 'x2': x2, 'y2': y2}.
19
20      Returns:
21          image_with_focused_x_values (PIL.Image.Image): the image with specified x values focused on
22
23      Example:
24          image = Image.open("sample_img.jpg")
25          image_with_focused_x_values = focus_on_x_values(image, ["2005", "2006"], {"2005": {'x1': 0.1, 'y1': 0.1, 'x2': 0.3, 'y2': 0.9}, "2006": {'x1': 0.4, 'y1': 0.1, 'x2': 0.6, '
    y2': 0.9}, "2007": {'x1': 0.7, 'y1': 0.1, 'x2': 0.9, 'y2': 0.9}})
26          display(image_with_focused_x_values)
27      """
28
29  def focus_on_y_values_with_mask(image, y_values_to_focus_on, all_y_values_bounding_boxes):
30      """
31      This function is useful when you want to focus on some specific y values in the image.
32      It does this by masking out the y values that are not needed.
33      This function is especially useful for horizontal bar charts.
34      For example, you can focus on the y values in a chart that are relevant to your analysis and ignore the rest.
35      Return the masked image.
36
37      Args:
38          image (PIL.Image.Image): the input image
39          y_values_to_focus_on (List[str]): a list of y values to focus on.
40          all_y_values_bounding_boxes (Dict[Dict]): a dictionary of bounding boxes for all y values in the image. key is y value and value is the bounding box of that y value. Each
    bounding box is in the format {'x1': x1, 'y1': y1, 'x2': x2, 'y2': y2}.
41
42      Returns:
43          image_with_focused_y_values (PIL.Image.Image): the image with specified y values focused on
44
45      Example:
46          image = Image.open("sample_img.jpg")
47          image_with_focused_y_values = focus_on_y_values(image, ["0", "10"], {"0": {'x1': 0.1, 'y1': 0.1, 'x2': 0.9, 'y2': 0.15}, "10": {'x1': 0.1, 'y1': 0.2, 'x2': 0.9, 'y2': 0.5},
    "20": {'x1': 0.1, 'y1': 0.6, 'x2': 0.9, 'y2': 0.9}})
48      """
49
50  def focus_on_x_values_with_draw(image, x_values_to_focus_on, all_x_values_bounding_boxes):
51      """
52      This function is useful when you want to focus on some specific x values in the image.
53      It does this by drawing a red box around the x values that need to be focused on.
54      This function is especially useful for vertical bar charts.
55      For example, you can focus on the x values in a chart that are relevant to your analysis.
56      Return the masked image.
57
58      Args:
59          image (PIL.Image.Image): the input image
60          x_values_to_focus_on (List[str]): a list of x values to focus on.
61          all_x_values_bounding_boxes (Dict[Dict]): a dictionary of bounding boxes for all x values in the image. key is x value and value is the bounding box of that x value. Each
    bounding box is in the format {'x1': x1, 'y1': y1, 'x2': x2, 'y2': y2}.
62
63      Returns:
64          image_with_focused_x_values (PIL.Image.Image): the image with specified x values focused on
65
66      Example:
67          image = Image.open("sample_img.jpg")
68          image_with_focused_x_values = focus_on_x_values(image, ["2005", "2006"], {"2005": {'x1': 0.1, 'y1': 0.1, 'x2': 0.3, 'y2': 0.9}, "2006": {'x1': 0.4, 'y1': 0.1, 'x2': 0.6, '
    y2': 0.9}, "2007": {'x1': 0.7, 'y1': 0.1, 'x2': 0.9, 'y2': 0.9}})
69          display(image_with_focused_x_values)
70      """
```

```
1   def focus_on_y_values_with_draw(image, y_values_to_focus_on, all_y_values_bounding_boxes):
2       """
3       This function is useful when you want to focus on some specific y values in the image.
4       It does this by drawing a red box around the y values that need to be focused on.
5       This function is especially useful for horizontal bar charts.
6       For example, you can focus on the y values in a chart that are relevant to your analysis.
7       Return the masked image.
8
9       Args:
10          image (PIL.Image.Image): the input image
11          y_values_to_focus_on (List[str]): a list of y values to focus on.
12          all_y_values_bounding_boxes (Dict[Dict]): a dictionary of bounding boxes for all y values in the image. key is y value and value is the bounding box of that y value. Each
    bounding box is in the format {'x1': x1, 'y1': y1, 'x2': x2, 'y2': y2}.
13
14      Returns:
15          image_with_focused_y_values (PIL.Image.Image): the image with specified y values focused on
16
17      Example:
18          image = Image.open("sample_img.jpg")
19          image_with_focused_y_values = focus_on_y_values(image, ["0", "10"], {"0": {'x1': 0.1, 'y1': 0.1, 'x2': 0.9, 'y2': 0.15}, "10": {'x1': 0.1, 'y1': 0.2, 'x2': 0.9, 'y2': 0.5},
    "20": {'x1': 0.1, 'y1': 0.6, 'x2': 0.9, 'y2': 0.9}})
20      """
21
22  def focus_on_x_values_with_highlight(image, x_values_to_focus_on, all_x_values_bounding_boxes):
23      """
24      This function is useful when you want to focus on some specific x values in the image.
25      It does this by adding light transparent red highlight to the x values that need to be focused on.
26      This function is especially useful for vertical bar charts.
27      For example, you can focus on the x values in a chart that are relevant to your analysis.
28      Return the masked image.
29
30      Args:
31          image (PIL.Image.Image): the input image
32          x_values_to_focus_on (List[str]): a list of x values to focus on.
33          all_x_values_bounding_boxes (Dict[Dict]): a dictionary of bounding boxes for all x values in the image. key is x value and value is the bounding box of that x value. Each
    bounding box is in the format {'x1': x1, 'y1': y1, 'x2': x2, 'y2': y2}.
34
35      Returns:
36          image_with_focused_x_values (PIL.Image.Image): the image with specified x values focused on
37
38      Example:
39          image = Image.open("sample_img.jpg")
40          image_with_focused_x_values = focus_on_x_values(image, ["2005", "2006"], {"2005": {'x1': 0.1, 'y1': 0.1, 'x2': 0.3, 'y2': 0.9}, "2006": {'x1': 0.4, 'y1': 0.1, 'x2': 0.6, '
    y2': 0.9}, "2007": {'x1': 0.7, 'y1': 0.1, 'x2': 0.9, 'y2': 0.9}})
41          display(image_with_focused_x_values)
42      """
43
44  def focus_on_y_values_with_highlight(image, y_values_to_focus_on, all_y_values_bounding_boxes):
45      """
46      This function is useful when you want to focus on some specific y values in the image.
47      It does this by adding light transparent red highlight to the y values that need to be focused on.
48      This function is especially useful for horizontal bar charts.
49      For example, you can focus on the y values in a chart that are relevant to your analysis.
50      Return the masked image.
51
52      Args:
53          image (PIL.Image.Image): the input image
54          y_values_to_focus_on (List[str]): a list of y values to focus on.
55          all_y_values_bounding_boxes (Dict[Dict]): a dictionary of bounding boxes for all y values in the image. key is y value and value is the bounding box of that y value. Each
    bounding box is in the format {'x1': x1, 'y1': y1, 'x2': x2, 'y2': y2}.
56
57      Returns:
58          image_with_focused_y_values (PIL.Image.Image): the image with specified y values focused on
59
60      Example:
61          image = Image.open("sample_img.jpg")
62          image_with_focused_y_values = focus_on_y_values(image, ["0", "10"], {"0": {'x1': 0.1, 'y1': 0.1, 'x2': 0.9, 'y2': 0.15}, "10": {'x1': 0.1, 'y1': 0.2, 'x2': 0.9, 'y2': 0.5},
    "20": {'x1': 0.1, 'y1': 0.6, 'x2': 0.9, 'y2': 0.9}})
63      """
64  ```
65  # GOAL #: Based on the above tools, I want you to reason about how to solve the # USER REQUEST # and generate the actions step by step (each action is a python jupyter notebook
    code block) to solve the request.
66  You may need to use the tools above to process the images and make decisions based on the visual outputs of the previous code blocks.
67  Your visual ability is not perfect, so you should use these tools to assist you in reasoning about the images.
68  The jupyter notebook has already executed the following code to import the necessary packages:
69  ```python
70  from PIL import Image
71  from IPython.display import display
72  from tools import *
73  ```
74
75  # REQUIREMENTS #:
76  1. The generated actions can resolve the given user request # USER REQUEST # perfectly. The user request is reasonable and can be solved. Try your best to solve the request.
77  2. The arguments of a tool must be the same format specified in # TOOL LIST #;
78  3. If you think you got the answer, use ANSWER: <your answer> Please extract the final answer in FINAL ANSWER: <final answer> and ends with TERMINATE.
79  4. All images in the initial user request are stored in PIL Image objects named image_1, image_2, ..., image_n. You can use these images in your code blocks. Use display() function
    to show the image in the notebook for you too see.
80  5. Use as few tools as possible. Only use the tools for the use cases written in the tool description. You can use multiple tools in a single action.
81  6. If you do not think you have enough information to answer the question on the images returned by the tools, you should directly answer the question based on the original image.
82  7. If all the x values are relevant to the question for a vertical bar chart, you do not need to focus on any specific x values. You should directly answer the question based on
    the original image.
83  8. If all the y values are relevant to the question for a horizontal bar chart, you do not need to focus on any specific y values. You should directly answer the question based on
    the original image.
84  Below are some examples of how to use the tools to solve the user requests. You can refer to them for help. You can also refer to the tool descriptions for more information.
```

```
1  # [ four in−context examples here ]
2
3  # USER REQUEST #:
```

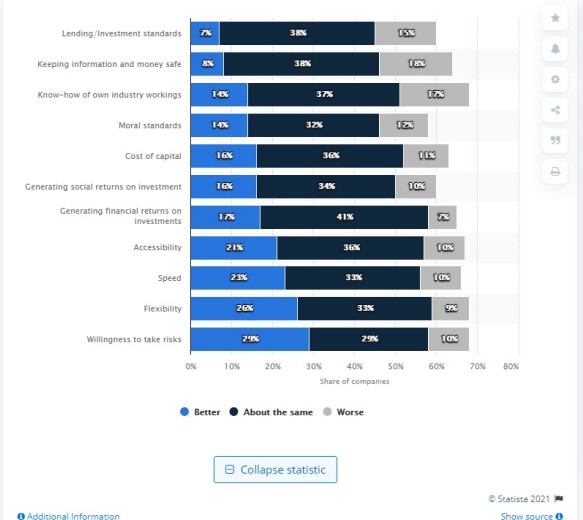

```
1  What is the average of blue bar above 20 value?
2
3  # USER Bounding Box Info: x_values_bbox, storing x values and coordinates . y_values_bbox, storing x values and coordinates . x_intervals_bbox , storing x intervals and coordinates .
   y_intervals_bbox , storing y intervals and coordinates . The x values in the image are :  []. The y values in the image are : [' Lending/Investment standards ', 'Keeping information and
   money safe ', 'Know−how of own industry workings', 'Moral standards ', 'Cost of capital ', 'Generating social returns on investment ', 'Generating financial returns on ', ' Accessibility
   ', ' Speed', ' Flexibility ', ' Willingness to take risks ']. The x intervals in the image are :  []. The y intervals in the image are : [].
4
5  # USER IMAGE stored in image_1, as PIL image.
6
7  Now please generate only THOUGHT 0 and ACTION 0 in RESULT. If no action needed, also reply with ANSWER: <your answer> Please extract the final answer in FINAL ANSWER: <final
   answer> and ends with TERMINATE:
8
9  # RESULT #:
```

### Thought + Action 0

```
1  THOUGHT 0: This is a horizontal bar chart image. I need to focus on the y values where the blue bar is above 20 and calculate the average of those values .
2
3  ACTION 0:
4  ```python
5  image_with_focused_y_values = focus_on_y_values_with_draw(image_1, [" Accessibility ", "Speed", " Flexibility ", "Willingness to take risks " ], y_values_bbox)
6  display (image_with_focused_y_values)
7  ```
```

### Observation 0

Execution success. The output is as follows.

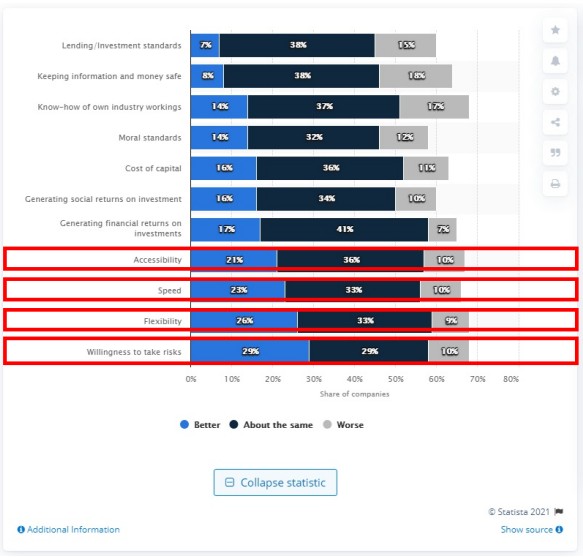

Answer the question What is the average of blue bar above 20 value? You can turn the table image into text and answer with step of thinking.
Reply with ANSWER: <your answer>
Please extract the final answer in FINAL ANSWER: <final answer> and ends with TERMINATE.

---

**Thought + Action 1**

---

ANSWER: The blue bars above 20 are for "Accessibility" (21%), "Speed" (23%), "Flexibility" (26%), and "Willingness to take risks" (29%). To find the average:

$$\text{Average} = \frac{21 + 23 + 26 + 29}{4} = \frac{99}{4} = 24.75$$

FINAL ANSWER: 24.75. TERMINATE

## D. Finetune Details

### D.1. REFOCUS Dataset Statistics

The detailed dataset statistics are demonstrated in Table 6. One example input data can be viewed in Listing 1.

|  | Horizontal Bar | Vertical Bar | Total |
|---|---|---|---|
| QA Data (Masry et al., 2022) | 4,990 | 10,069 | 15,059 |
| REFOCUS Data | 4,722 | 9,622 | 14,344 |
| w/ Editing | 4,220 | 8,599 | 12,819 |

*Table 6.* **Detailed statistics about REFOCUS Data.** We count the total number of our training cases, and the ones with visual editing.

### D.2. Finetune Experiment Details

For the fine-tuning experiments in Section 5, we use 8 NVIDIA RTX A6000 GPUs (48GB RAM per GPU), following the Phi-3.5-vision github instructions[4]. We adopt full training, and iterate through the hyperparameter searching space to find the best performing set to report as our result score. The hyperparameter search focuses specifically on (1)learning rate, (2) epoch number, (3) whether to include edited image in training input, with learning rate ranging in (5e-5 5e-6 5e-4 1e-4 1e-5 1e-6 5e-7), epoch ranging in (1 2), and all other parameters set to default values. In Table 7, we present detailed hyper-parameter configurations for results in Table 4.

|  | w/ default QA Data | w/ REFOCUS VCoT | w/ REFOCUS CoT |
|---|---|---|---|
| data type | bf16 | bf16 | bf16 |
| batch size | 64 | 64 | 64 |
| learning rate | $5 \times 10^{-7}$ | $1 \times 10^{-6}$ | $5 \times 10^{-6}$ |
| epoch number | 2 | 2 | 2 |
| include edited image in input | No | No | Yes |

*Table 7.* Hyper-parameter settings for our best fine-tuned models.

### D.3. SFT Result Analyses

Looking at the SFT results in Table 4, we see a consistent improvement on the chart problems using REFOCUS data. If we use the de facto QA data to finetune, the chain of thought ability will be impaired, comparing to the original model. Also, the improvement on Vertical Bar problems reaches 5.4%, which is quite large for 14k training data.

## E. SFT Qualitative Examples

In this section, we show some example output from the Phi-3.5-vision model finetuned on REFOCUS visual CoT data in Figure 8. Notice that the model output are in text formats with bounding box coordinates for the areas to focus on. We draw these areas in red boxes for illustration purposes. The examples are selected from the Horizontal Bar set and Vertical Bar set.

---

[4]https://github.com/microsoft/Phi-3CookBook/blob/main/md/04.Fine-tuning/FineTuning_Vision.md

**Listing 1** Example REFOCUS data input for SFT experiment. The "response0" and "response1" are GPT-4o outputs that generate codes and answer based on the edited image respectively. We use "vcot_input" as the target answer.

```
 1   {
 2       "id": "train-two_col_103562",
 3       "query": "As of 2021, how many championship titles had Ferrari won?",
 4       "answer": "16",
 5       "source": "h_bar",
 6       "images": ["two_col_103562.png"],
 7       "response0": "This is a horizontal bar chart image. I need to focus on the part where
 8       the y-axis value is "Ferrari" to find out how many championship titles they have won.",
 9       "edited_images": ["two_col_103562/f19f2fd043444680a02764d66fd6b22d.png"],
10       "response1": "As of 2021, Ferrari had won 16 championship titles.",
11       "focus_areas": [
12           {
13               "x1": 5,
14               "y1": 38,
15               "x2": 795,
16               "y2": 72
17           }],
18       "vcot_input": "This is a horizontal bar chart image. I need to focus on the part where
19       the y-axis value is "Ferrari" to find out how many championship titles they have won.
20       The areas to focus on in the image have bounding box coordinates: [{"x1": 5, "y1":
21       38, "x2": 795, "y2": 72}]. Looking at these areas, As of 2021, Ferrari had won
22       16 championship titles."
23   }
```

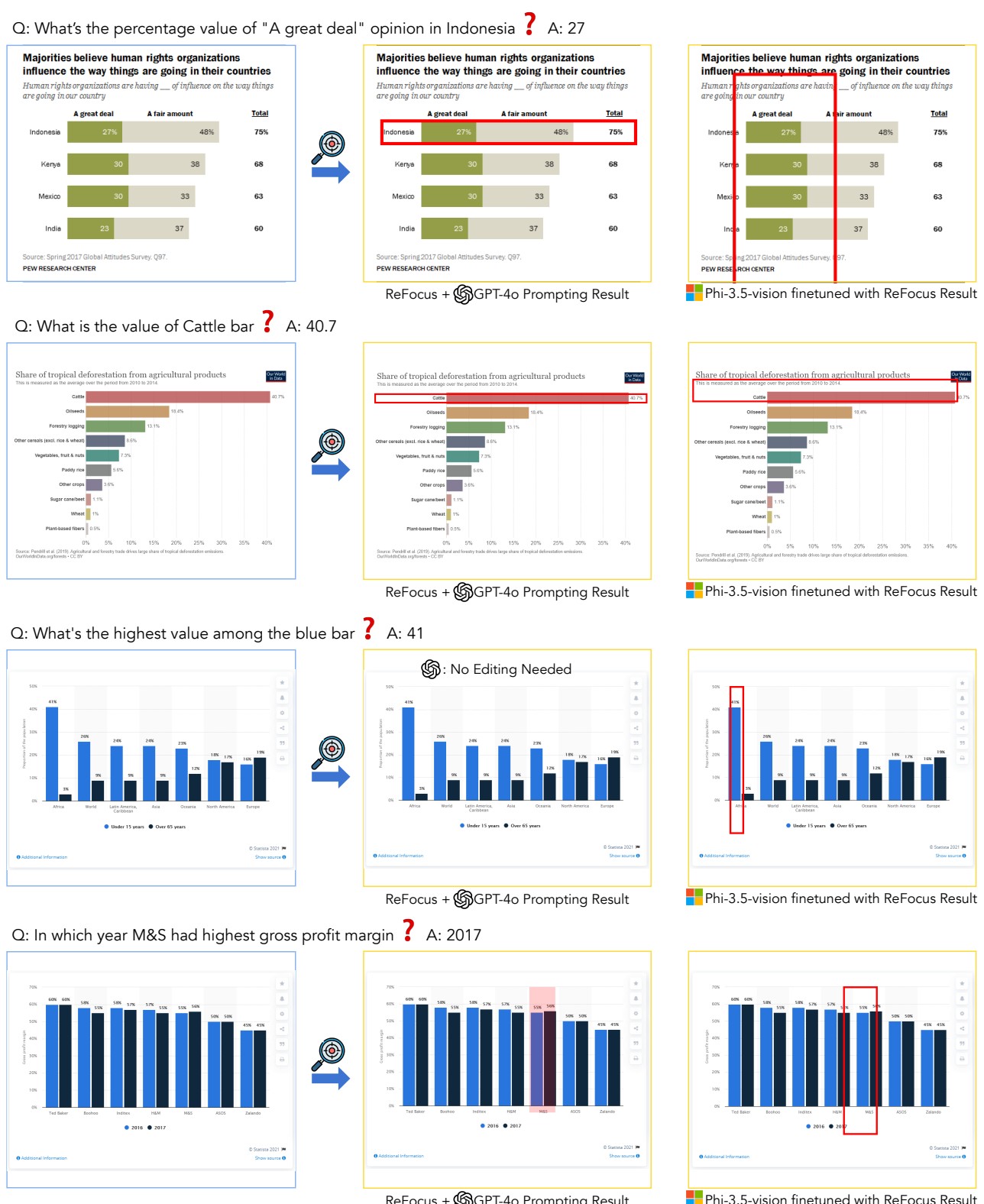

Q: What's the percentage value of "A great deal" opinion in Indonesia ❓ A: 27

Q: What is the value of Cattle bar ❓ A: 40.7

Q: What's the highest value among the blue bar ❓ A: 41

Q: In which year M&S had highest gross profit margin ❓ A: 2017

*Figure 8.* **Phi-3.5-vision finetuned with REFOCUS visual chain of thought data** outputs the areas to focus on. For illustration purposes, we draw these areas in red boxes, and compare with the REFOCUS + GPT-4o prompting output.

