# OpenReview forum: "ReFocus: Visual Editing as a Chain of Thought for Structured Image Understanding"
_ICML.cc/2025/Conference — ICML 2025 poster_

### Official Review · Reviewer_TSQ2 · 2025-03-06

**Overall Recommendation:** 2

**Summary:**

This paper introduces a visual editing method where authors proposed a range of tools (i.e., python functions) that vision language models can use to edit the image (e.g., highlight, draw box, mask out) to better focus on the structured visual content corresponding to the text query. Experiments show that this process improves model performance on certain tabular and chart understanding tasks, and authors show that open-weight vision language models can also benefit from this process by being finetuned on data that involves tool calling that leads to the correct answer.

## update after rebuttal
I would like to thank the authors for their rebuttal and thoughtful follow-up addressing my concerns and questions. I have carefully re-evaluated the manuscript, particularly Section 5, and respectfully disagree with the authors’ claim that their method demonstrates the ability of models to generate novel functions for visual editing at test time (which was the core of my initial question). In my view, providing function definitions as part of the prompt, or generating answers in the same structured format as used during training (e.g., `<image> <question> <thought1> <ReFocus bounding box> <thought2> <answer>`), does not constitute novel test-time function generation. Rather, it reflects learned behavior from training patterns, not evidence of the model inventing new visual editing logic on its own. Because authors did not show that models can generate novel visual editing functions, the answer to the follow-up question "can models make good use of these self-generated functions to edit the image and improve their performance" remains unclear.

That said, I do agree with the authors that the proposed prompting approach is effective, and that fine-tuning models with rejection-sampled visual CoT data distilled from trajectories does lead to improved performance. However, I still remain hesitant in considering the work to be above the bar for a conference paper due to the limited scope and a relative lack of novel technical contributions.

**Claims And Evidence:**

- I think that the claims are well-supported in general, though the comparison with Visual Sketchpad that it incorporated external expert knowledge while ReFocus does not require additional information does not appear to be fair. In particular, ReFocus still relies on algorithmic tools (i.e., `findContours` , `getStructuringElement` ) to extract coordinates of table/chart element, which weakens the argument that it doesn’t rely on additional information. To really support this claim, an ideal setting would be that the model needs to provide the coordinates as input arguments (instead of providing any labels to mask and let the algorithm decide the region).
- For your claim that multimodal models lack selective attention and multi-hop visual reasoning ability mentioned in the introduction, can authors provide supporting literature and/or additional experiments?

**Essential References Not Discussed:**

n/a

**Experimental Designs Or Analyses:**

On the analysis that discussed the particular editing method that works the best, authors should constrain their conclusion toward that particular dataset for a particular model. The conclusion might not be generalizable on other datasets or other models.

**Methods And Evaluation Criteria:**

Yes, the selection of evaluation dataset appears to be suitable within its application scope.

**Other Comments Or Suggestions:**

n/a

**Other Strengths And Weaknesses:**

- The paper is clearly written, which I really appreciate.
- I think the contribution is too incremental and limited in scope. In particular, I believe ReFocus is simply an analogy to Visual Sketchpad used for structured images (with a different set of tools). Because all the functions are pre-defined and the functionality is hardcoded, the experiments can only be done with very few datasets that show compatibility with these handcrafted tools. It’s not a general method that improves vision language model on a wide range of datasets, not even to say applications. I would be much more convinced if you can show that models can generate code on-the-fly to achieve some “focus” functionality and apply it to a variety of structured image evaluations instead of sticking to a few hand-crafted tools on very few evaluation settings.

**Questions For Authors:**

- I’m curious if you can provide any cost / speed analysis in inference because apparently I think there’s a trade-off between better accuracy and prolonged generation in thoughts and IO.

**Relation To Broader Scientific Literature:**

The general finding aligns with Visual Sketchpad where visual editing improves vision language model performance.

**Theoretical Claims:**

n/a

---

> ### Author Rebuttal · Authors · 2025-04-01
>
> We appreciate the reviewer's valuable feedback. Below, we provide a detailed response to address the concerns and questions raised:
>
> * Q1. External Expert Knowledge Comparison with Visual SketchPad
>
> We would like to clarify some confusions here. What we want to say is that, the tools used in ReFocus, do not bring in external information – every information is constrained to what’s provided in the input images. However, in Visual SketchPad, specialists such as Depth Anything will bring in external information (e.g. a depth map)  to the process, to assist in decision making. We'll ensure this distinction is clearer in our revisions.
>
> * Q2. Claim about Lack of multi-hop visual reasoning ability
>
> Basically, we were referring to all different kinds of multimodal LLMs (e.g. GPT, Gemini, Claude, LLaVa, LLaVa-Next, etc.), when we ask a VQA question, they could not answer while providing the selective attention areas or multi-hop visual reasoning processes. When we play with these models, they would at most generate some thinking processes in text, but not in visual format, on structured images, as we explained in the introduction. Many related literature are listed in Related Work starting line 154. But we were just making a very general claim through our observations, that multimodal LLMs cannot return the visual area to focus on during question answering.
>
> * Q3. Analysis Constraints
>
> We will update in our revisions to constrain our conclusions toward a particular dataset for a particular model when discussing a particular editing method that works the best.
>
>
> * Q4. Contribution & Comparison with Visual SketchPad
>
> We would like to clarify any confusion regarding this concern. ReFocus is the first method to excel at structured image tasks, which Visual SketchPad cannot solve. **More importantly, ReFocus explores whether it can provide a more human-like heuristic supervision signal and if this supervision is practically useful—unlike Visual SketchPad, which is limited to prompting only.** In our fine-tuning experiments (Section 5), we found that training data obtained through ReFocus is 8% more effective than default VQA training pairs and 2.6% better than Chain-of-Thought data. We believe our fine-tuned model aligns with the reviewer’s definition of “performing on-the-fly refocus without relying solely on a few tools”.
>
>
> * Q5. Cost / Speed Trade-off Analysis
>
> Due to lack of time, we conducted a model usage analysis on Chart Horizontal Bar Set.
> As for the consideration of speed, the generated token count with ReFocus is about twice as that of the baseline model.
> As for the consideration of cost, the total token count is approximately ten times higher with ReFocus than the baseline. This can be too costly for API-based models like GPT-4o. In this context, our fine-tuned model offers a practical alternative. While it may not be as effective as ReFocus combined with GPT-4o, it is cost-free.
> We will include these discussions in our revisions.

---

> > ### Comment · Reviewer_TSQ2 · 2025-04-03
> >
> > Thanks for the authors' responses! I understand the notion of bringing external information or not better. But I am still not sure if it's the right term to use. While a depth map is definitely external information, I would argue that ReFocus also brings external information, such as visual cues that are not originally present in the image that are helpful for model to localize areas of interests. The distinction definitely has to be made clearer.
> >
> > I appreciate the authors' response to Q3. But it further echoed one of my original concerns that the framework is limited in scope. The current formulation of ReFocus is not a general solution for structured image understanding in terms of image-editing based CoT. Like the authors' mentioned in Q4, the framework first *manually* crafts a few human-like heuristics in terms of functions and performs SFT on models, so that the models can learn these functions and generate the corresponding code calling *these functions* during inference.
> >
> > That said, one cannot easily scale the number of functions with the framework, or ways in visual heuristics that the models can employ. To really show the generality of this framework, I would expect that, instead of having humans to write the functions for models to learn and use during inference time, *can models write the functions themselves during inference depending on the questions for the visual cues that they want to apply (e.g., what if the model wants to draw a horizontal line aligned to the top of a bar across all bars)?* For example, I can see that in page 13 and 14 where you give the model all the contexts for the functions that you designed, and in page 15 the model answers by calling such functions i.e., `focus_on_columns_with_draw`. My question toward this framework has two parts: (1) can models generate new functions on their own instead of relying on what are provided by humans; and (2) can models make good use of these self-generated functions to edit the image and improve their performance. This is different from the authors' current paradigm, and explains my previous comment on *“performing on-the-fly refocus without relying solely on a few tools”*.
> >
> > Connecting this disadvantage all the way back to limitation in scope -- I am sure that there will be structured image scenarios where all the currently manually designed functions will be useless, and in that case models that learn to use these functions will eventually fail, but models that learn to synthesize new functions that are tailored to the scenarios and make correct uses of their generated functions will eventually likely to succeed.
> >
> > Authors' response in Q5 addressed my corresponding question.
> >
> > Given the authors' responses and the scope/applicability of the work, I maintain my rating as a weak rejection.

---

> > > ### Author Response · Authors · 2025-04-05
> > >
> > > We appreciate the reviewer's feedback!
> > >
> > > * External Information
> > >
> > > As for the distinction of external information -- we will make it clearer in our revisions!
> > >
> > > * Scope Limitation
> > >
> > > We agree that the prompting part is limited in scope. However, as we show in the Finetune experiments (Section 5), **once we use ReFocus to collect training data and finetune a better model using this data, we are no longer limited to tools or prompts to realize the purpose of "refocus"**.
> > >
> > > For the questions:
> > >
> > > (1) can models generate new functions on their own instead of relying on what are provided by humans: --
> > >
> > > Yes, not need any tools or prompting, we realize a general function which is "providing refocus area", through our finetuning approach;
> > >
> > > (2) can models make good use of these self-generated functions to edit the image and improve their performance: --
> > >
> > > Yes, in the prompting experiments, the editing functions that we provided based on the idea of "focus on certain area" can improve multimodal LLMs performance.
> > >
> > > (3) (e.g., what if the model wants to draw a horizontal line aligned to the top of a bar across all bars)? --
> > >
> > > This is an extension of what editing function can help models better solve the problems. It would be interesting to explore the answer, but our current prompting experiments with provided editing functions already support our findings (which is that refocus helps), and our finetuning experiments show a general way to turn these specific editing functions into general refocusing ability.
> > >
> > > We hope this additional response can help explain things better. Please let us know if reviewer has further questions. Thanks!

---

### Official Review · Reviewer_r3TB · 2025-03-09

**Overall Recommendation:** 3

**Summary:**

This paper proposes ReFocus, a visual reasoning framework that enhances structured image understanding with editing on the input image. Specifically, ReFocus iteratively highlights some columns and rows in the tabular problems, masks the other information, and draws bounding boxes on the information that needs attention on. Experimental results show that the proposed method ReFocus yields consistent performance gain across various tasks.

## Update after rebuttal
The rebuttal addresses most of my concerns about calling the model "chain-of-thought", inference time comparison, etc. One remaining issue that I have is the potential insights in the proposed method that I find hard to evaluate on. The proposed method still seems to be some prompt engineering plus some simple tricks, for me as a person mainly working on general computer vision. Editing something by highlighting [1] to improve performance is also not a brand new technique in computer vision. However, due to my limited knowledge on structured image understanding, I want to request for AC and other reviewers' expertise on this field to tell me whether the proposed method is insightful enough to be accepted to ICML, or it is just some simple tricks and prompt engineering. Therefore, I want to flag this potential issue for AC to confirm the novelty and insight in the proposed method before this paper gets accepted.

[1] Shtedritski et al. What does CLIP know about a red circle? Visual prompt engineering for VLMs. ICCV 2023.

**Claims And Evidence:**

Yes, the claim that the proposed ReFocus can help structured image understanding (more specifically, tabular problems) is well validated by the experimental results across multiple benchmarks across tabular understanding tasks.

**Essential References Not Discussed:**

I do not find essential related works that are not discussed. However, this may not be accurate because I am not very familiar with the field of structured image understanding.

**Experimental Designs Or Analyses:**

The experimental designs and analyses are sound and valid to me, and also comprehensive as it is conducted across multiple benchmarks and evaluated on state-of-the-art multi-modal LLMs like many versions of GPT-4o.

**Methods And Evaluation Criteria:**

Yes, the proposed method makes sense, and I think it is intuitive as it follows the way we humans process the tabular data, as we first focus on several rows / columns that we are interested in, while ignoring other irrelevant rows / columns, then reasoning across the relevant information in the tabular data.

**Other Comments Or Suggestions:**

I think Section C in the supplementary material is important for understanding how the proposed method works. When I only read Section 3 in the main paper, actually I cannot understand how the proposed method actually works. The information of adopting the "iterative chain-of-thoughts (although in the Weaknesses part I said I think calling it "chain-of-thoughts" may be overclaiming) until reaching TERMINATE" strategy for conducting the method is important for understanding how the algorithm works. Instead, the main paper spend much space on explaining the different structured image understanding tasks, which I think is unnecessary and can be moved to the supplementary.

**Other Strengths And Weaknesses:**

**Strengths:**
- The proposed method ReFocus conforms to the reasoning process of humans, which looks intuitive to me.
- The experiments seem to be comprehensive, covering a broad range of benchmarks and structured image understanding tasks and shows the superior performance after incorporating ReFocus on the existing powerful multi-modal LLMs like GPT-4o.

**Weaknesses:**
- I think it is a bit overclaiming to call the proposed method "chain-of-thought". Essentially, the proposed method is iterate the columns / rows to highlight until the multi-modal LLMs determines TERMINATE. Basically, there is no specific "chain-of-thought" from the multi-modal LLM agent to find the next most possible rows / columns in a heuristic way by itself with reasoning. Therefore, I think the "reasoning process" within this paper is more primitive than the one in the original paper [1], which may be overclaiming to call it "chain-of-thought".

- **(I am doubtful of whether it should be a strength / weakness, but I am leaning towards weakness right now)** I often in favor of a method that is "simple but effective". For the method proposed in this paper, I think it is effective as supported by the experimental results. However, I think it seems to be too simple to me (because it is basically just iterating the rows and columns to highlight some of them until reaching TERMINATE to get the final answer), which seems more like a simple trick to me, instead of providing insights in the scientific / techincal parts.

[1] Wei et al. Chain-of-Thought Prompting Elicits Reasoning in Large Language Models. NeurIPS 2022.

**Questions For Authors:**

- The main reason for me to give a weak reject as the initial score is that I think the proposed method is so simple that it seems to be a trick for performance improvement instead of providing scientific / techincal insights. What do the authors think of my concerns and arguments? What additional insights the authors want to convey to me about the proposed method?
- Since the proposed method needs to iterate through the rows and columns to highlight some of them. How is the inference time after adding ReFocus to the multi-modal LLMs compared to the original base models?

**Relation To Broader Scientific Literature:**

The capability of multi-modal LLMs to understand structured images like charts and figures is broadly related to the advancement towards artificial general intelligence.

**Theoretical Claims:**

The paper has no theoretical claims.

---

> ### Author Rebuttal · Authors · 2025-04-01
>
> We appreciate the reviewer's valuable feedback. Below, we provide a detailed response to address the concerns and questions raised:
>
> * Q1. Confusion about “Chain-of-Thought” and “Iterations”
>
> We would like to clarify any confusion regarding this concept. **ReFocus does not iteratively highlight every column or row.** Instead, it leverages a multimodal LLM to determine which columns or rows to highlight or maskout by generating Python code. For instance, in Figure 1, when provided with an image, the multimodal LLM identifies focus areas (Column “Country”, Column “Wins”) and generates Python code `focus_on_columns_with_maskout(“Country”, “Wins”)` (where  `focus_on_columns_with_maskout` is one of the edit functions, and all edit function names are provided to the LLM in prompt as in Supplementary C. the LLM decides which function to use). The generated edit function is then executed through our backend support. Next, the LLM assesses whether additional edits are necessary (because as in Fig1 it needs two edits), and if so it decides autonomously on the next focus areas. (And this is where the word iteration comes from.) This method allows models to perform self-reason, justifying the use of the term "chain-of-thought." We add "visual" to CoT to reflect the presence of intermediate visual editing outputs.
>
> * Q2. Simplicity of the Method
>
> The difficulty of this method during the prompting stage is specifically how to enable the method to self-reason about what to focus on, rather than relying on iterations. We are the first paper to demonstrate that using simple tools and editing can achieve this goal on structured images.
>
> **Another challenge is determining whether ReFocus can provide a more human-like heuristic supervision signal and whether this supervision is useful in practice.** In our fine-tuning experiments, we found that the training data obtained through ReFocus is 8% more effective than default VQA training pairs and 2.6% better than Chain-of-Thought training data. Overall, both the prompting and fine-tuning goals present significant challenges.
>
> * Q3. Inference Time
>
> Naturally, the inference time increases when incorporating ReFocus compared to using the original base models. It can be seen as roughly twice the inference time, transitioning from Question -> Answer, to Question -> Edit Decisions -> Answer. We did check the model usage, and the generated token number with ReFocus is about twice as that of baseline model.
>
> Another consideration is cost. For example, on the Chart Horizontal Bar set, the total token count is approximately ten times higher with ReFocus than the baseline setting. This can be too costly for API-based models like GPT-4o. In this context, our fine-tuned model offers a practical alternative. While it may not be as effective as ReFocus combined with GPT-4o, it is cost-free.
>
> We hope our answers can address the reviewer’s concerns! We’re happy to provide further clarification if needed.

---

> > ### Comment · Reviewer_r3TB · 2025-04-03
> >
> > Dear authors,
> >
> > Thanks for your response! My comments are as follows:
> >
> > The authors have clarified that the multimodal LLMs will decide which row/column to focus on, instead of iterating every row/column. Now I agree that it is somewhat mimicing the chain-of-thought process. But I do think that the organization of the paper is suboptimal. I think the unnecessarily long introduction of all the "Structured Image Problems" needs to be moved to the supplementary, while the much more important method explanation needs to be emphasized and explained more clearly in the main paper. For the inference time, I am OK with it being twice long as the original model, because naturally more chain-of-thought increases the inference time, but presenting this in the paper could be crucial to audience. For the simplicity of the method, this is something that I still hold reservations. The proposed method still seems to be some prompt engineering plus some simple tricks, for me as a person mainly working on general computer vision. Editing something by highlighting [1] to improve performance is also not a brand new technique in computer vision. However, due to my limited knowledge on structured image understanding, I want to request for AC and other reviewers' expertise on this field to tell me whether the proposed method is insightful enough to be accepted to ICML, or it is just some simple tricks and prompt engineering. Therefore, **I want to flag this potential issue for AC to confirm the novelty and insight in the proposed method before this paper gets accepted**.
> >
> > Based on the above information, I can raise my score to 3, but **I would like to request judgement for the novelty of the proposed method from AC and other reviewers that have more expertise**.
> >
> > [1] Shtedritski et al. What does CLIP know about a red circle? Visual prompt engineering for VLMs. ICCV 2023.

---

> > > ### Author Response · Authors · 2025-04-04
> > >
> > > We appreciate the reviewer's feedback and score adjustment! We will optimize paper organization in our revisions!
> > >
> > > As for the contribution of the paper, we want to emphasize that ReFocus is **not only a prompting paper**, because it proves to be a **better supervision signal** as in our fine-tuning experiments (Section 5). Basically, we (1) first show that prompting works, and (2) this prompting way can allow us collect better training data to train better vision language models.
> > >
> > > Please let us know if this further clarifies reviewer's confusion. We will for sure include all the additional response in our revisions.

---

### Official Review · Reviewer_phJH · 2025-03-11

**Overall Recommendation:** 3

**Summary:**

This paper focuses on the structured image tasks, using the Python codes to edit the input image as the "visual thought" and the modified image can be more focused on the target information of the question, which benefits the accuracy of the answer.

**Claims And Evidence:**

Yes

**Essential References Not Discussed:**

No.

**Experimental Designs Or Analyses:**

1. For the analysis of Refocus to improve grounding and OCR capabilities, it is better to add more experimental results instead of giving conclusions through only two examples.
2. Lack of analysis for some results, like Table 2.

**Methods And Evaluation Criteria:**

The evaluation datasets seem small, why only a subset is selected for evaluation and what is the standard to select data.

**Other Comments Or Suggestions:**

No

**Other Strengths And Weaknesses:**

Strengths
1. The motivation of this paper is very clear.
2. The collected 14k training set is effective.
Weaknesses
1. The method is limited and can only be applied to the chart that has a specific structure.
2. The method is more like introducing a strong prior specifically for certain data, thereby enhancing performance.
3. The method exhibits significant performance fluctuations across different models, and even different versions of the same GPT model can show substantial variations. Additionally, the performance may have adverse effects on some models, and the author lacks discussion on these points. Meanwhile, please also see the 'Experimental Designs Or Analyses'.

**Questions For Authors:**

Why was the current evaluation dataset selected? For the bar chart, the other datasets like DVQA have more scales of test data and can provide a more effective evaluation. The current testing scale is too small, and the test results also show significant performance fluctuations between different models and datasets. This may stem from the unreliable evaluations caused by the small scale. Meanwhile, due to the strong limitations and priors of the method, the limited data scale can not evaluate the extensibility of the method well.

**Relation To Broader Scientific Literature:**

This method proposed a visual CoT approach which uses Python codes as intermediary.

**Theoretical Claims:**

There's no theoretical claim.

---

> ### Author Rebuttal · Authors · 2025-04-01
>
> We appreciate the reviewer's valuable feedback. Below, we provide a detailed response to address the concerns and questions raised:
>
>
> * Q1. Evaluation Dataset Selection.
>
> The major reason for testing on ChartQA instead of on DVQA is that DVQA is a synthetic (chart and data) dataset with template questions, while ChartQA and CharXiv include real charts and data with open-form questions that have been manually checked. Additionally, ChartQA has been much more widely tested by many recent models (as in Gemini and GPT reports) than DVQA so that it’s easier to compare with them, and CharXiv has its unique challenges of scientific chart reading.
>
>
> * Q2. Evaluation Set Size Concerns
>
> We want to emphasize that small scale evaluation data does not mean unreliable evaluations. For example, datasets like Winoground (CVPR 2022), which contains only 400 examples, have significantly influenced VLM advancements. Similarly, although focused on different aspects, other widely adopted benchmarks—including WHOOPS! (ICCV 2023), LlaVA-Bench (NeurIPS 2023), Visit-Bench (NeurIPS 2024), ConTextual (ICML 2024), VibeEval, and Visual Riddles (NeurIPS 2024)—comprise 90, 500, 576, 500, 269, and 400 examples, respectively, and have been pivotal for evaluating VLMs. Our total evaluation set has a size of 2219 instances, and should reflect a reliable evaluation result.
>
>
> * Q3. Performance Fluctuations between different models.
>
> Yes, there are performance differences between different multimodal LLMs, but we believe the Table1 results show relatively consistent improvement. The programing ability and the visual prompt understanding ability of LLMs will both affect the effectiveness of ReFocus. However, we want to emphasize that (1) ReFocus is a simple yet effective general pipeline that can work with different multimodal LLMs, and (2) we add the fine-tuning experiments to reduce the reliance on prompting to a certain degree.
>
>
> * Q4.  Limitations and Priors of the method
>
> We are not sure if we understand what the prior is as in “The method is more like introducing a strong prior specifically for certain data”. Could the reviewer kindly elaborate more on what the “prior” is? If the reviewer means that the method favors structured image data, then we want to emphasize that ReFocus was designed for solving this kind of problems with simple editing processes. Extending to other types of image data is for sure interesting and worth exploring, but a bit beyond our paper’s scope.

---

> > ### Comment · Reviewer_phJH · 2025-04-03
> >
> > Thanks for the author's feedback.
> > 1. I agree that small scale evaluation data does not mean unreliable evaluations, but I also believe that a larger scale is more reliable and I don't find it difficult to enlarge the test set scale in this paper's case.
> > 2. I still think this method has significant limitations due to its requirement for structured data, which limits the impact of this paper.
> >
> > Based on the above reasons, I will keep my original score. Thanks.

---

> > > ### Author Response · Authors · 2025-04-04
> > >
> > > We appreciate the reviewer's feedback. We will try to make the scale larger in the future, Thanks!

---

### Official Review · Reviewer_cQeo · 2025-03-11

**Overall Recommendation:** 4

**Summary:**

The paper proposed a LMM + tool-using style framework for structured image understanding. Specifically, the framework provides tools to edit the original image and either highlight the important region or remove irrelavent region. Results show that this framework generally improves LMMs zero-shot performance on understanding tables, charts, etc. The paper additionally show that using the framework to curate SFT data provides better supervision signal for these tasks.

## Update after rebuttal
The rebuttal provides additional infomation, I am keeping my already positive evaluation

**Claims And Evidence:**

The claims are well-scoped and the results supports the claims.

Additional explanation/analysis are needed; see the Experimental Designs Or Analyses section.

**Essential References Not Discussed:**

N/A

**Experimental Designs Or Analyses:**

The experiments are mostly solid and informative;

Additional anaysis is needed in:
- It's good to show error analysis on the tools and how it will be propagated to the final performance;
- Categorize what are the remaining errors that cannot be addressed even with accurate visual editing;
- For some categories, ReFocus fail to yield better performance, e.g., VTabFact in Table2; What's the reason? Additional analysis would be helpful

**Methods And Evaluation Criteria:**

The evaluation setting makes sense; The strength is the paper also includes finetuning experiments;

**Other Comments Or Suggestions:**

N/A

**Other Strengths And Weaknesses:**

The paper is clearly written and easy to follow

**Questions For Authors:**

N/A

**Relation To Broader Scientific Literature:**

This is in line with research in multimodal reasoning on abstract visual concepts.

**Theoretical Claims:**

N/A

---

> ### Author Rebuttal · Authors · 2025-04-01
>
> We appreciate the reviewer's insightful feedback. Below, we provide a detailed response to address the concerns and questions raised:
>
> * Q1. Tool Errors
>
> We carefully examined this phenomena through manual checking, on around 40 data cases each for Table VWTQ and Chart Horizontal Bar datasets.
> For table editing, most of them (roughly 80%) are correct. For the problematic ones, there are two big error types. The main error (\~80% of which) is that sometimes, both column editing and row editing are needed, but only one of them was applied, making the original problem, e.g. counting specific rows, a bit easier but not easy enough for the model to solve. A small portion (\~20% of which) is that it could mask out important information (e.g. some rows or columns)  that should not be masked out.
> For chart editing, it’s similar that most of them(\~90%) are correct. The main error is that sometimes the python program is incorrect, and the wrong area gets edited, e.g. the chart should focus on the last three bars, but the last 4th instead of the last 3rd bar was highlighted, therefore the final calculation uses the incorrect number of the bar, etc.
>
> * Q2. Remaining Errors if Visual Editing is Accurate
>
> Some complicated reasoning cases still remain a challenge. For instance, if a chart image has three bars, and all of them needs to be considered, compared, and calculated with each other, then even if ReFocus highlights all three bars, it still makes the same mistake as if no edit is performed.
>
> * Q3. Performance Difference.
>
> For VTabFact in Table2, we believe it is because Phi-3-vision may not deal with visual prompts that well, compared to dealing with the original image, because it was largely trained on OCR tasks and has seen many original text-rich images. We believe that the programing ability of the multimodal LLM and the visual prompt understanding ability will both affect the effectiveness of ReFocus.

---

> > ### Comment · Reviewer_cQeo · 2025-04-03
> >
> > Thank authors for providing the rebuttal, I keep my already positive rating

---

> > > ### Author Response · Authors · 2025-04-04
> > >
> > > We appreciate the reviewer's feedback and will include our additional response in our revisions.

---

### Decision · Program_Chairs · 2025-05-01

**Decision:**

Accept (poster)

**Comment:**

This paper presents ReFocus, a method that lets vision-language models edit images (like highlighting or masking) to improve structured image understanding tasks such as tables and charts. The idea is to mimic visual chain-of-thought by letting the model focus step by step. Reviewers generally liked the paper, and AC also likes the idea. It is clearly written and shows good performance gains. One strength is that ReFocus not only helps prompting but also provides better training data, improving model fine-tuning. Reviewers' concerns span from small evaluation datasets and limited generality, but authors addressed these with a diligent attempt, showing comparisons to related works and discussing future extensions. AC also feel calling it “chain-of-thought” might be too strong, but after author clarification, I feel it is also okay... AC feels having a logo in front of the title is disturbing, and may cause paper parsing engines to introduce errors for archiving, please consider removing it.

Overall, this is a useful and well-presented paper with practical impact of interest to ICML community. I recommend a weak accept.